# Autoinhibition of Munc18-1 modulates synaptobrevin binding and helps to enable Munc13-dependent regulation of membrane fusion

Ewa Sitarska[1,2,3†‡], Junjie Xu[1,2,3†], Seungmee Park[4], Xiaoxia Liu[1,2,3], Bradley Quade[1,2,3], Karolina Stepien[1,2,3], Kyoko Sugita[4], Chad A Brautigam[1,5], Shuzo Sugita[4], Josep Rizo[1,2,3]*

[1]Department of Biophysics, University of Texas Southwestern Medical Center, Dallas, United States; [2]Department of Biochemistry, University of Texas Southwestern Medical Center, Dallas, United States; [3]Department of Pharmacology, University of Texas Southwestern Medical Center, Dallas, United States; [4]Department of Physiology, University of Toronto, Toronto, Canada; [5]Department of Microbiology, University of Texas Southwestern Medical Center, Dallas, United States

*For correspondence: Jose.Rizo-Rey@UTSouthwestern.edu

†These authors contributed equally to this work

Present address: ‡Cell Biology and Biophysics Unit, European Molecular Biology Laboratory, Heidelberg, Germany

Competing interests: The authors declare that no competing interests exist.

**Abstract** Munc18-1 orchestrates SNARE complex assembly together with Munc13-1 to mediate neurotransmitter release. Munc18-1 binds to synaptobrevin, but the relevance of this interaction and its relation to Munc13 function are unclear. NMR experiments now show that Munc18-1 binds specifically and non-specifically to synaptobrevin. Specific binding is inhibited by a L348R mutation in Munc18-1 and enhanced by a D326K mutation designed to disrupt the 'furled conformation' of a Munc18-1 loop. Correspondingly, the activity of Munc18-1 in reconstitution assays that require Munc18-1 and Munc13-1 for membrane fusion is stimulated by the D326K mutation and inhibited by the L348R mutation. Moreover, the D326K mutation allows Munc13-1-independent fusion and leads to a gain-of-function in rescue experiments in *Caenorhabditis elegans unc-18* nulls. Together with previous studies, our data support a model whereby Munc18-1 acts as a template for SNARE complex assembly, and autoinhibition of synaptobrevin binding contributes to enabling regulation of neurotransmitter release by Munc13-1.

## Introduction

Neurotransmitter release is crucial for interneuronal communication. This exquisitely regulated process involves the docking of synaptic vesicles at presynaptic active zones, priming of the vesicles to a readily-releasable state and very fast (<1 ms) $Ca^{2+}$-triggered fusion of the vesicle and plasma membranes (*Südhof, 2013*). The machinery that controls release contains core proteins that form part of a universal intracellular membrane fusion apparatus. These proteins include (*Rizo and Xu, 2015*; *Südhof and Rothman, 2009*; *Jahn and Fasshauer, 2012*): (i) the synaptic vesicle SNAP receptor (SNARE) synaptobrevin and the plasma membrane SNAREs syntaxin-1 and SNAP-25, which together form a tight four-helix bundle called the SNARE complex that brings the two membranes together and is key for membrane fusion (*Söllner et al., 1993*; *Hanson et al., 1997*; *Poirier et al., 1998*; *Sutton et al., 1998*); (ii) N-ethylmaleimide-sensitive factor (NSF) and soluble NSF attachment proteins (SNAPs; no relation to SNAP-25), which disassemble the SNARE complex to recycle the SNAREs (*Söllner et al., 1993*; *Mayer et al., 1996*; *Banerjee et al., 1996*); and (iii) the Sec1/Munc18

**eLife digest** Nerve cells communicate with other nerve cells by releasing small molecules called neurotransmitters. The neurotransmitters are first packaged inside bubble-like structures called vesicles, which fuse with the membrane of the nerve cell when it is stimulated. Once the vesicle and membrane have fused, the neurotransmitters are released outside the nerve cell and are detected when they bind to proteins on the surface of other nearby nerve cells.

A machinery of different proteins controls membrane fusion. Amongst these proteins are five called Munc18-1, Munc13-1, syntaxin-1, synaptobrevin and SNAP-25. The last three form a tight bundle called SNARE complex that brings the vesicle and cell membrane together and is essential for the two to fuse. Munc18-1 and Munc13-1 orchestrate the assembly of the SNARE complex. Previous studies suggested that Munc18-1 binds to synaptobrevin, providing a template to bring syntaxin-1 and synaptobrevin together and thereby helping the SNARE complex to form. However, the importance of the interaction between Munc18-1 and synaptobrevin was not clearly established, and it was not known how Munc13-1 is involved.

Sitarska, Xu et al. have now measured how mutated versions of Munc18-1 bind to synaptobrevin and tested how the mutations affect membrane fusion. A mutation in Munc18-1 that increased binding to synaptobrevin increased membrane fusion too, while a mutation that decreased binding had the opposite effect and reduced fusion. The results support the idea that Munc18-1 provides a template for the SNARE complex to form. One mutation stimulated Munc18-1 so that Munc13-1 was no longer needed for fusion when the mutant Munc18-1 was tested in fusion assays with artificial membranes. This mutation was designed to perturb the structure of a region of Munc18-1 protein that normally inhibits the binding of synaptobrevin.

These results suggest that by adopting a state where it cannot bind synaptobrevin, Munc18-1 can only be stimulated to form the SNARE complex and trigger release of neurotransmitter when Munc13-1 is present. This provides a way for Munc13-1, which is regulated by many factors, to fine-tune the release of neurotransmitter. Future work will test whether these proteins work in the same way in living animals. This will help us understand how communication between neurons is finely controlled to enable the brain to carry out its many different tasks.

(SM) protein Munc18-1/UNC-18, which (together with Munc13s/UNC13) orchestrates SNARE complex formation in an NSF-SNAP-resistant manner (*Ma et al., 2011*, *2013*). In addition, the neuronal exocytotic machinery contains multiple additional proteins, such as the $Ca^{2+}$ sensor Synaptotagmin-1, RIMs, Rab3s, CAPS and complexins, which confer the tight regulation of neurotransmitter release (*Rizo and Xu, 2015*).

This tight regulation also arises in part from the core components themselves. Thus, Munc13s contain a MUN domain (*Basu et al., 2005*) that has homology to tethering factors from diverse membrane compartments (*Pei et al., 2009*; *Li et al., 2011*) and hence provides a core function, but also multiple domains with key regulatory roles (e.g. *Lu et al. [2006]*; *Deng et al. [2011]*; *Liu et al. [2016]*). Syntaxin-1 also modulates neurotransmitter release by forming a self-inhibited 'closed conformation' that involves intramolecular binding of its N-terminal $H_{abc}$ domain to its SNARE motif, and that binds tightly to Munc18-1 (*Dulubova et al., 1999*; *Misura et al., 2000*). Opening of syntaxin-1 to form the SNARE complex is mediated by the Munc13 MUN domain (*Ma et al., 2011*; *Yang et al., 2015*). This transition constitutes a central step that can be regulated via Munc13s in presynaptic plasticity processes that mediate diverse forms of information processing in the brain (*Rizo and Rosenmund, 2008*). However, the mechanism of syntaxin-1 opening and the exact role of Munc18-1 in SNARE complex assembly have been enigmatic.

Some evidence suggested that the Munc13-1 MUN domain opens syntaxin-1 through a weak interaction with its SNARE motif, providing a template for SNARE complex assembly (*Ma et al., 2011*). However, an alternative model postulated that Munc18-1 provides a template for the formation of the SNARE complex. This model is based in part on the findings that Munc18-1 binds to synaptobrevin (*Xu et al., 2010b*) and that a Munc18-1 mutation that disrupts this interaction (L348R) impairs the stimulation of liposome lipid mixing by Munc18-1 in reconstitution assays

(*Parisotto et al., 2014*). Tantalizing evidence supporting a general role for SM proteins as templates for SNARE complex assembly was provided by two crystal structures of Vps33, the SM protein involved in yeast vacuolar fusion, bound either to Vam3 or to Nyv1, the respective homologues of syntaxin-1 and synaptobrevin in this system (*Baker et al., 2015*). Superposition of the two structures clearly shows that binding to the SM protein places the two SNARE motifs in close proximity and in the right register to form the SNARE complex (*Figure 1A*). Munc18-1 is likely to template SNARE complex assembly through similar interactions: the crystal structure of closed syntaxin-1 bound to Munc18-1 (*Misura et al., 2000*) revealed an orientation of the SNARE motif similar to that observed for Vam3 bound to Vps33 (compare *Figure 1A and B*). Furthermore, L348 side chain implicated in synaptobrevin binding (*Parisotto et al., 2014*) is located in a groove between two helices of domains 3a of Munc18-1 (*Figure 1C*), which is involved in the binding of Vps33 to Nyv1 (*Figure 1A*).

Despite this convergence of results, the nature of the synaptobrevin-Munc18-1 interaction remains unclear, as it is very weak and appears to be primarily mediated by the C-terminal region of the synaptobrevin SNARE motif (*Xu et al., 2010b*). This C-terminal region includes the sequence

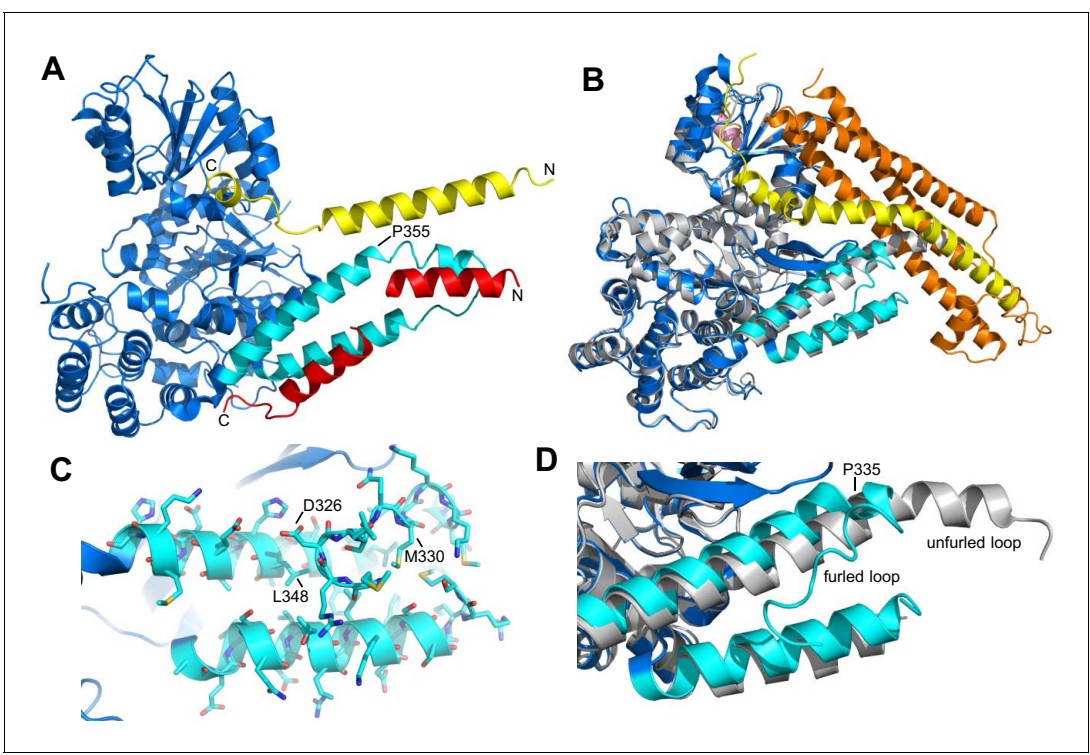

**Figure 1.** Autoinhibition by a loop of Munc18-1 is likely to inhibit binding to synaptobrevin. (**A**) Ribbon diagram of the structure of Vps33 (blue) bound to the Nyv1 SNARE motif (red) (PDB code 5BV0) superimposed with the structure of Vps33 (not shown) bound to the Vam3 SNARE motif (yellow) (PDB code 5BUZ) (*Baker et al., 2015*). The two helices connected by a loop in domain 3a of Vps33, which are involved in Nyv1 binding, are shown in cyan. (**B**) Crystal structure of Munc18-1 (blue) bound to closed syntaxin-1 (SNARE motif in yellow; N-terminal region in orange) (PDB code 3C98) (*Burkhardt et al., 2008*) superimposed to the crystal structure of Munc18-1 (gray) bound to a syntaxin-4 N-terminal peptide (pink) (PDB code 3PUJ) (*Hu et al., 2011*). The helix-loop-helix of domain 3a of Munc18-1 in the complex with syntaxin-1 is in cyan. (**C**) Close-up of the helix-loop-helix of domain 3a of Munc18-1 bound to syntaxin-1 (not shown) illustrating how the loop forms a 'furled conformation' by folding back onto the groove between the two helices that is putatively involved in synaptobrevin binding. Side chains are shown as stick models and those that were mutated to disrupt synaptobrevin binding are labeled. (**D**) Close-up of the helix-loop-helix region from the superposition shown in (**B**) but without showing syntaxin-1. Note how in the complex with closed syntaxin-1, the Munc18-1 loop is furled (cyan). whereas in the complex with the syntaxin-4 N-terminal peptide, the loop is unfurled and forms a helical extension of one of the helices but with a bend (gray). This bend has P335 in the corner, and the homologous residue of Vps33 (P355) is also in the corner of the helix bend (**A**).

The following figure supplement is available for figure 1:

**Figure supplement 1.** Sequence alignment of the helix-loop-helix region of domain 3a of rat Munc18-1 (residues 298–355) with the corresponding sequences of Munc18-1 from different species and of Vps33 from *Chaetomium thermophilum*.

KLKRKYWWK (residues 83–91), which contains five basic and three aromatic residues and is thus likely to be promiscuous, as shown by the finding that this region also binds to the Munc13-1 MUN domain (see below). Its homologous region in Nyv1 is not involved in Vps33 binding (*Baker et al., 2015*). The weak affinity of Munc18-1 for synaptobrevin may arise because of autoinhibition by a loop that connects the two aforementioned helices of domain 3a of Munc18-1, which occludes the putative synaptobrevin binding site in the crystal structure of Munc18-1 bound to closed syntaxin-1, forming what is referred to as a 'furled loop conformation' (*Figure 1B,D*) (*Hu et al., 2011*). Interestingly, this loops adopts an 'unfurled conformation' in the crystal structure of Munc18-1 bound to a syntaxin-4 N-terminal peptide (*Figure 1B,D*) (*Hu et al., 2011*), forming an extension of one of these helices that resembles the conformation observed for the homologous sequence in the Vps33-Nyv1 structure (*Figure 1A*). Thus, a conformational change in this loop may modulate synaptobrevin binding. Indeed, a P335A mutation designed to favor this helical extension enhanced Munc18-1 activity in lipid mixing assays (*Parisotto et al., 2014*) and caused a gain-of-function in secretory cells (*Han et al., 2014*; *Munch et al., 2016*). However, the basis for these results is unclear because the mutation did not appear to affect synaptobrevin binding (*Parisotto et al., 2014*), and this region does not form a continuous helix in Vps33 bound to Nyv1; rather, it forms a bent helix with a proline in the corner (P355, homologous to Munc18-1 P335; see *Figure 1A*). Furthermore, we do not yet know how the Munc18-1-synaptobrevin interaction is related to the functional interplay between Munc18-1 and Munc13s in orchestrating SNARE complex assembly.

In the study presented here, we addressed these questions using biophysical approaches and genetic experiments in *C. elegans*. NMR data show that the synaptobrevin SNARE motif binds to Munc18-1 through two types of interactions, one that involves the C-terminal region of the synaptobrevin SNARE motif and is likely non-specific, and another involving more central synaptobrevin sequences that is specifically impaired by the L348R mutation and enhanced by a mutation (D326K) designed to disrupt the furled loop conformation of Munc18-1. Using reconstitution assays in which liposome fusion strictly requires Munc18-1 and a Munc13-1 C-terminal fragment (*Ma et al., 2013*; *Liu et al., 2016*), thus mirroring the total abrogation of neurotransmitter release observed in the absence of Munc18-1 or Munc13s (*Verhage et al., 2000*; *Richmond et al., 1999*; *Aravamudan et al., 1999*; *Varoqueaux et al., 2002*), we find that the L348R mutation strongly impairs membrane fusion whereas the D326K mutation leads to a dramatic gain-of-function that can partially overcome the requirement for Munc13-1 for membrane fusion. Furthermore, the Munc18-1 D326K mutant rescues physiological defects observed in *C. elegans unc-18* nulls much more efficiently than WT Munc18-1, thus demonstrating a gain-of-function in a live animal. Overall, our data support the proposal that Munc18-1 acts as a template for the assembly of the SNARE complex by binding not only to syntaxin-1 but also to synaptobrevin. Moreover, our results suggest that autoinhibition of Munc18-1 binding to synaptobrevin enables or facilitates the existence of Munc13-dependent modes of regulation of neurotransmitter release, adding to the notion that diverse intra- and intermolecular inhibitory interactions within the release machinery provide opportunities for multiple layers of regulation that are fundamental for brain function.

## Results

### NMR analysis of the Munc18-1 binding sites in synaptobrevin

In previous cross-linking experiments, we found that a sequence from the C-terminus of the synaptobrevin SNARE motif was cross-linked with a sequence from the loop in domain 3a of rat Munc18-1 (*Xu et al., 2010b*). Because rat Munc18-1 tends to aggregate, we used NMR spectroscopy in that study to analyze the binding of synaptobrevin to squid Munc18-1, which is more soluble, and also found that binding was dominated by the C-terminus of the synaptobrevin SNARE motif (*Xu et al., 2010b*). In subsequent studies of interactions of neuronal SNAREs with Munc13-1, we found that the C-terminus of the synaptobrevin SNARE motif also binds to the Munc13-1 MUN domain via its C-terminus (*Figure 2—figure supplement 1*). These findings raised the possibility that this C-terminal region of the synaptobrevin SNARE motif is promiscuous, which is not surprising because it includes five basic and three aromatic residues. This sequence is close to the transmembrane region, and is therefore likely to interact with the synaptic vesicle membrane through its abundant basic and

hydrophobic residues in vivo; however, in the absence of membranes, this sequence is expected to be prone to non-specific protein interactions.

These observations, and the finding that the homologous sequence at the C-terminus of the Nyv1 SNARE motif does not bind to Vps33 (*Baker et al., 2015*), led us to examine the binding of synaptobrevin to Munc18-1 in more detail using NMR spectroscopy. For this purpose, we used a $^{15}$N-labeled fragment that spans the cytoplasmic region of synaptobrevin [synaptobrevin(1-96)] and acquired $^{1}$H-$^{15}$N HSQC spectra in the absence and presence of rat Munc18-1 at moderate concentrations (14.5 μM), which limited Munc18-1 aggregation while maintaining high sensitivity. Note that these spectra contain one cross-peak for each non-proline residue in a $^{15}$N-labeled protein, and that the intensity of the cross-peaks should decrease strongly upon binding to another protein (*Rizo et al., 2012*), particularly considering that the synaptobrevin cytoplasmic region is unstructured in isolation (*Hazzard et al., 1999*). Our $^{1}$H-$^{15}$N HSQC spectra confirmed the binding of Munc18-1 to the C-terminus of synaptobrevin(1-96), but showed that additional sequences from the SNARE motif also bind to Munc18-1 (*Figure 2*). The finding that the cross-peaks do not disappear completely indicates that the interaction is weak but, interestingly, examination of the cross-peak intensity ratios in the presence and absence of Munc18-1 shows clear intensity decreases not only for the C-terminal region (beyond residue 82) but also for residues 60–80; even residues 42–55 exhibit some meaningful intensity decreases albeit smaller. Note that the absolute intensities and relative intensity patterns in these experiments are highly reproducible, as illustrated by comparison of absolute intensities from two $^{1}$H-$^{15}$N HSQC spectra acquired for two different samples of $^{15}$N-synaptobrevin(1-96) plus Munc18-1 (*Figure 2—figure supplement 2*), and by the similarity of intensity ratio patterns observed in spectra acquired with WT and mutant Munc18-1s in regions where the mutations did not affect binding (see below and *Figure 3*).

It is also worth noting that, based on the homology between synaptobrevin and Nyv1 and on the crystal structure of Vps33 bound to Nyv1 (*Figure 1A*), an analogous binding mode for the neuronal proteins would predict that residues 41–55 of synaptobrevin bind to Munc18-1 at the loop region containing the helical extension, while residues 59–78 of synaptobrevin bind to the groove between the two helices of domains 3a. This prediction fits approximately with the decreases in cross-peak intensities observed in our $^{1}$H-$^{15}$N HSQC spectra, particularly considering that the smaller intensity decreases observed for residues 42–55 (*Figure 2B*) may arise because the binding of this region may be hindered by competition with the C-terminus, which interacts with the same region of Munc18-1 according to the cross-linking experiments (see above). In summary, these results suggest that the Vps33-Nyv1 binding mode is conserved for neuronal Munc18-1 and synaptobrevin, and that analysis of this interaction by methods that do not provide residue-specific information is complicated by the existence of non-specific binding involving the C-terminus of the synaptobrevin SNARE motif. We attempted to use $^{1}$H-$^{15}$N HSQC spectra to analyze Munc18-1 binding to a $^{15}$N-labeled synaptobrevin fragment spanning residues 1–82, which lacks the promiscuous C-terminal region, but no cross-peaks could be observed for the SNARE motif of this fragment because of aggregation.

## Specific alteration of synaptobrevin binding by mutations in domain 3a of Munc18-1

Previous biochemical data suggested that the L348R mutation disrupts binding of Munc18-1 to synaptobrevin and impairs the activity of Munc18-1 in liposome lipid mixing assays (*Parisotto et al., 2014*). To design additional mutations that could potentially correlate synaptobrevin binding to Munc18-1 function, we examined which residues of Munc18-1 might be involved in binding based on homology with the Vps33-Nyv1 structure, and which side chains may stabilize the furled conformation of the loop in domain 3a of Munc18-1 (*Figure 1C*), as destabilization of this conformation may enhance synaptobrevin binding. These analyses were hindered by the low sequence similarity between Munc18-1s and Vps33 (*Figure 1—figure supplement 1*), and because the furled conformation of the Munc18-1 loop is not very well defined in the crystal structure of its complex with syntaxin-1. (Note that the conformation of the loop is different in the original published structure (*Misura et al., 2000*) and that refined later with the same data [*Burkhardt et al., 2008*]). Nevertheless, D326 appeared to be a good candidate that may stabilize the furled loop because it may participate in hydrogen-bonding interactions (*Figure 1C*) and is highly conserved throughout evolution in Munc18-1s (*Figure 1—figure supplement 1*). Conversely, M330 may contribute to synaptobrevin

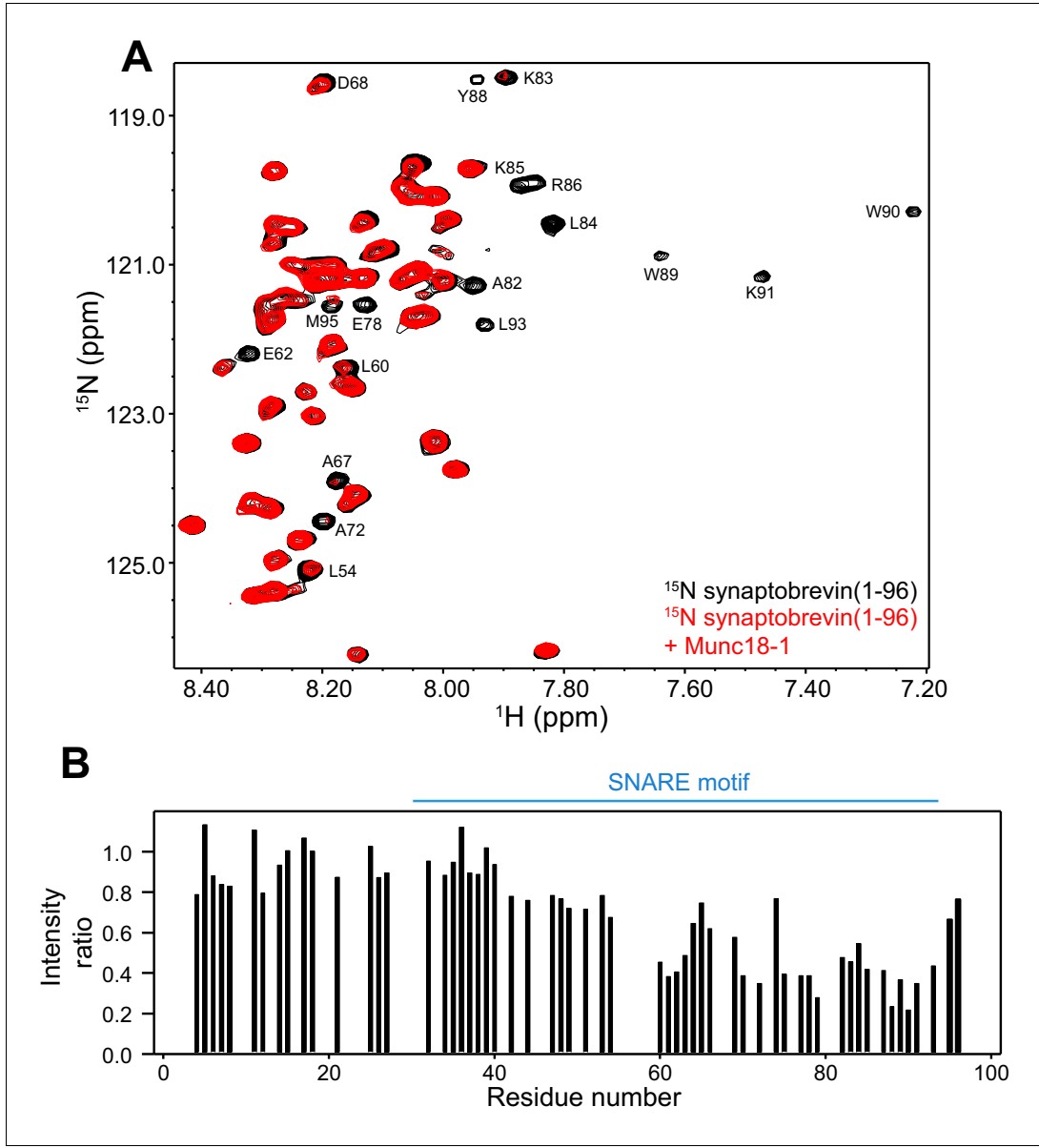

**Figure 2.** Munc18-1 binds to central sequences and the C-terminus of the synaptobrevin SNARE motif. (**A**) $^{1}$H-$^{15}$N HSQC spectra of 14.5 μM $^{15}$N-synaptobrevin(1-96) in the absence (black contours) and presence (red contours) of 14.5 μM Munc18-1. Selected cross-peaks that were particularly broadened by Munc18-1 binding are labeled. (**B**) Plots of the ratios of the intensities of the synaptobrevin(1-96) $^{1}$H-$^{15}$N HSQC cross-peaks observed in the presence of Munc18-1 versus those observed in its absence, as a function of residue number. Intensities were measured only for well-resolved cross-peaks. Ratios were calculated from experiments performed with two separate samples and averaged. Note that the synaptobrevin SNARE motif spans approximately residues 29–91 (***Sutton et al., 1998***).

The following figure supplements are available for figure 2:

**Figure supplement 1.** The Munc13-1 MUN domain binds to the C-terminus of the synaptobrevin SNARE motif but not to more central sequences.

**Figure supplement 2.** Reproducibility of $^{1}$H-$^{15}$N HSQC cross-peak intensities.

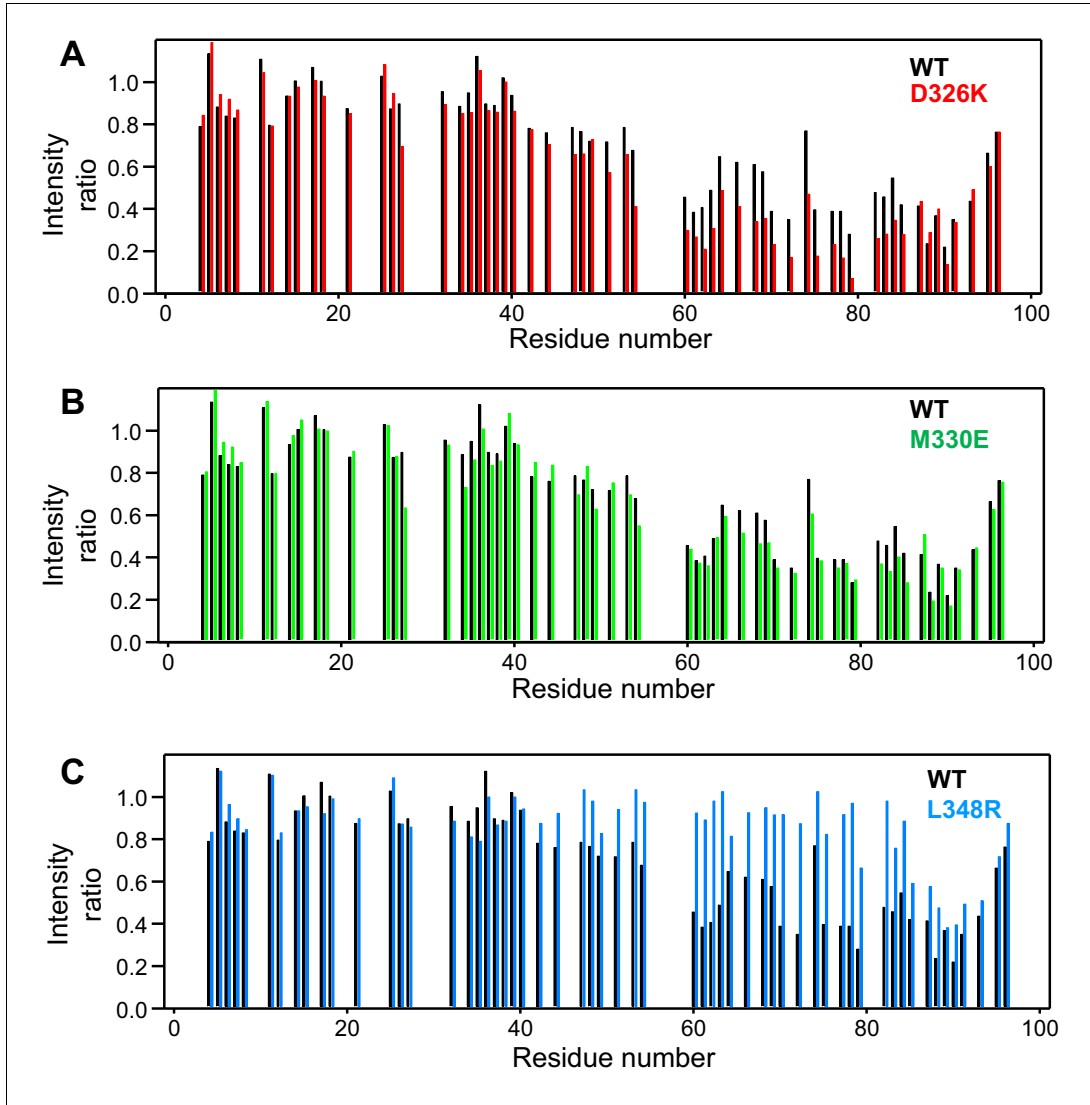

**Figure 3.** Selective enhancement and disruption of synaptobrevin binding by Munc18-1 mutations. (**A–C**) Plots of the ratios of the intensities of the synaptobrevin(1-96) $^1$H-$^{15}$N HSQC cross-peaks observed in the presence of WT or mutant Munc18-1 versus those observed in its absence, as a function of residue number. The plots compare the results obtained with WT Munc18-1 (black bars) with those obtained with the D326K (red bars, [**A**]), M330E (green bars, [**B**]) or L348R (blue bars, [**C**]) Munc18-1 mutants. Intensities were measured only for well-resolved cross-peaks. Ratios were calculated from experiments performed with two separate samples and averaged.

binding because it is in a conserved hydrophobic position and its homologous residue of Vps33 participates in binding to Nyv1.

We used $^1$H-$^{15}$N HSQC spectra to assess whether a mutation that reverses the charge of D326 (D326K) in Munc18-1 or a mutation that disrupts the hydrophobicity of M330 (M330E) could alter binding to $^{15}$N-synaptobrevin(1-96), and also analyzed the effects of the L348R mutation. Importantly, the D326K Munc18-1 mutant caused stronger decreases in the intensities of cross-peaks from residues 45–83 than those caused by WT Munc18-1 (*Figure 3A*). These effects seem moderate, but they do reflect a clear increase in the specific binding of the D326K mutant to this region. By contrast, the patterns of intensity ratios resulting from the addition of the D326K mutant or WT Munc18-1 are very similar for other regions of synaptobrevin, including the 40 N-terminal residues that are not involved in binding and the C-terminal region, which does bind to Munc18-1 but through interactions that are not affected by the D326K mutation. The M330E Munc18-1 mutant

caused very similar intensity changes on the synaptobrevin $^1$H-$^{15}$N HSQC cross-peaks to those caused by WT Munc18-1 (*Figure 3B*). Hence, the M330 side chain does not contribute substantially to the affinity of Munc18-1 for synaptobrevin, but the M330E mutation provides a convenient negative control as a mutation in the loop region that is predicted to have no or very limited functional effects. Conversely, the L348R Munc18-1 mutant caused practically no decreases in the intensities of the synaptobrevin $^1$H-$^{15}$N HSQC cross-peaks except for those from the C-terminus, where the changes exhibited a similar pattern to those caused by WT Munc18-1. These data correlate with previous biochemical results showing impaired binding of the L348R Munc18-1 mutant to synaptobrevin (*Parisotto et al., 2014*). Moreover, these results provide an emphatic illustration of the power of this method of analysis of protein–protein interactions, showing that the L348R mutation specifically impairs binding of Munc18-1 to the sequences of the synaptobrevin SNARE motif that are expected to bind according to the crystal structure of the Vps33–Nyv1 complex, and that the mutation has a clearly less marked effect on the non-specific binding involving the C-terminus.

Overall, these results support the notion that there is a specific interaction between Munc18-1 and synaptobrevin that resembles that observed between Vps33 and Nyv1, and that this interaction is autoinhibited by the furled conformation of the loop in domain 3a of Munc18-1. We also analyzed whether the D326K, M330E and L348R mutations alter the binding of Munc18-1 to syntaxin-1 or to the SNARE complex using isothermal titration calorimetry (ITC) (*Table 1*). The $K_D$s measured for the binding of WT and mutant Munc18-1s to syntaxin-1 were all very low, refining to less than 2 nM. The differences in the $K_D$s calculated in duplicate experiments for each Munc18-1 protein and the confidence intervals derived from the fits (*Table 1*) indicate that none of the differences in $K_D$s measured for WT Munc18-1 and the mutants can be considered significant. Hence, these data show that the three Munc18-1 mutants retain very high affinity for syntaxin-1, although we cannot rule out the possibility that the mutations alter the affinity to some extent that is not measurable in these experiments. Similarly, there do not appear to be major differences in the affinities of WT and mutant Munc18-1s for the SNARE complex, although it is plausible that the D326K mutant has an increased affinity. Because Munc18-1 is relatively unstable (*Xu et al., 2011*), we also measured the thermal denaturation curves of the three Munc18-1 mutants using circular dichroism. We observed no significant differences in the melting temperature compared to WT Munc18-1, indicating that the mutations do not destabilize the protein. Therefore, the functional consequences of the D326K, M330E and L348R mutations in Munc18-1 are not expected to arise from misfolding.

## Strong activation of Munc18-1 by the D326K mutation

Previous analyses of the effects of the L348R mutation on the activity of Munc18-1 in stimulating lipid mixing between reconstituted SNARE proteoliposomes were very insightful but did not include Munc13, NSF or SNAPs (*Parisotto et al., 2014*), and hence cannot connect these effects to the interplay between Munc18-1, Munc13, NSF and SNAPs in SNARE complex assembly ordisassembly. To study the functional consequences of the D326K, M330E and L348R mutations in Munc18-1, we used assays that monitor both lipid and content mixing (*Zucchi and Zick, 2011*) between liposomes containing synaptobrevin (V-liposomes) and liposomes containing syntaxin-1-SNAP-25 (T-liposomes) in the presence of different combinations of NSF, αSNAP, Munc18-1, a fragment spanning the two C$_2$ domains of Synaptotagmin-1 (C$_2$AB) and a fragment spanning the conserved C-terminal region of Munc13-1 (i.e. containing its C$_1$, C$_2$B, MUN and C$_2$C domains, referred to as C$_1$C$_2$BMUNC$_2$C) (*Liu et al., 2016*). Monitoring of content mixing is necessary to ascertain that real membrane fusion occurs, as extensive lipid mixing can occur without content mixing (*Chan et al., 2009*; *Zick and Wickner, 2014*; *Liu et al., 2016*). The inclusion of NSF and αSNAP is critical to render membrane fusion strictly dependent on both Munc18-1 and Munc13 (*Ma et al., 2013*), mirroring the total abrogation of neurotransmitter release observed in their absence (*Verhage et al., 2000*; *Richmond et al., 1999*; *Aravamudan et al., 1999*; *Varoqueaux et al., 2002*).

All experiments were started in the absence of Ca$^{2+}$, and Ca$^{2+}$ was added after 300 s to study how it affects membrane fusion. As expected, control experiments performed in the presence of NSF-αSNAP revealed very little fusion between V- and T-liposomes in the absence of Ca$^{2+}$, and highly efficient fusion upon Ca$^{2+}$ addition when both WT Munc18-1 and Munc13-1 C$_1$C$_2$BMUNC$_2$C were included, but not when either one or both of these proteins were absent (*Figure 4A,B*). Interestingly, in analogous experiments that included Munc13-1 C$_1$C$_2$BMUNC$_2$C and Munc18-1 mutants,

**Table 1.** Summary of the ITC data obtained to analyze the binding of WT and mutant Munc18-1s to syntaxin-1 and the SNARE complex[*].

| Syringe | Cell | Cell concentration correct factor[†] | $K_D$ [nM] | ΔH[kcal/Mol] |
|---|---|---|---|---|
| Syx2-253 | WT-1 | 0.98 [0.97, 0.98] | 0.10 [0, 0.15] | −22.80 [−23.13, −22.46] |
| | WT-2 | 0.92 [0.92, 0.93] | 0.34 [0, 0.67] | −22.03 [−22.36, −21.71] |
| | D326K-1 | 0.89 [0.89, 0.89] | 0.23 [0.11, 0.37] | −23.56 [−23.75, −23.37] |
| | D326K-2 | 0.86 [0.85, 0.87] | 0.30 [0, 0.67] | −25.45 [−26.30, −24.61] |
| | M330E-1 | 0.92 [0.91, 0.94] | 1.76 [0.82, 3.23] | −21.56 [−22.21, −20.92] |
| | M330E-2 | 0.97 [0.94, 1.00] | 1.52 [0.49, 3.47] | −25.67 [−27.43, −24.04] |
| | L348R-1 | 0.86 [0.85, 0.87] | 0.54 [0.23, 0.98] | −25.42 [−25.90, −24.95] |
| | L348R-2 | 0.85 [0.85, 0.85] | 0.71 [0.51, 0.93] | −25.12 [−25.31, −24.92] |
| SNARE complex[‡] | WT-1 | 1.07 [0.90, 1.18] | 1,382 [952, 2172] | −5.21 [−7.31, −4.24] |
| | WT-2 | 1.01 [0.84, 1.11] | 1,479 [1018, 2355] | −5.46 [−7.99, −4.38] |
| | D326K-1 | 1.05 [0.98, 1.12] | 442 [296, 681] | −4.29 [−4.94, −3.84] |
| | D326K-2 | 0.98 [0.81, 1.11] | 361 [179, 834] | −3.13 [−4.66, −2.51] |
| | M330E-1 | 0.96 [0.79, 1.08] | 730 [414, 1459] | −2.48 [−3.72, −1.99] |
| | M330E-2 | 0.95 [0.82, 1.12] | 1,205 [993, 1462] | −3.69 [−4.30, −3.25] |
| | L348R-1 | 0.95 [0.89, 1.00] | 677 [521, 895] | −5.06 [−5.74, −4.56] |
| | L348R-2 | 0.89 [0.77, 0.98] | 781 [527, 1245] | −5.20 [−7.00, −4.30] |

[*]Two independent experiments were performed for WT Munc18-1 and for each mutant. For all parameters, 68.3% confidence intervals calculated using the error-surface projection method are indicated between square brackets.

[†]Correction factor calculated as part of the fitting procedure to account for differences in the concentrations of active proteins (**Brautigam et al., 2016**).

[‡]The SNARE complex was formed with syntaxin-1 (2–253), synaptobrevin(1–96), SNAP-25(11-82) and SNAP-25(141–203).

we found that fusion was strongly impaired by the L348R mutation, which disrupts binding of Munc18-1 to synaptobrevin, whereas the D326K mutation that enhances synaptobrevin binding caused a dramatic enhancement of fusion before $Ca^{2+}$ addition, and fusion was not affected by the M330E mutation that does not alter synaptobrevin binding (**Figure 4C,D**; see quantification of content mixing after 150 s and 500 s in **Figure 4—figure supplement 1A,B**). These results establish a clear correlation between the activity of the Munc18-1 mutants in supporting membrane fusion in this reconstituted system and their ability to bind to synaptobrevin.

As expected, the Synaptotagmin-1 $C_2AB$ fragment did not have major effects in these experiments (**Figure 4—figure supplement 1C,D**) because the $Ca^{2+}$-dependence of fusion arises primarily

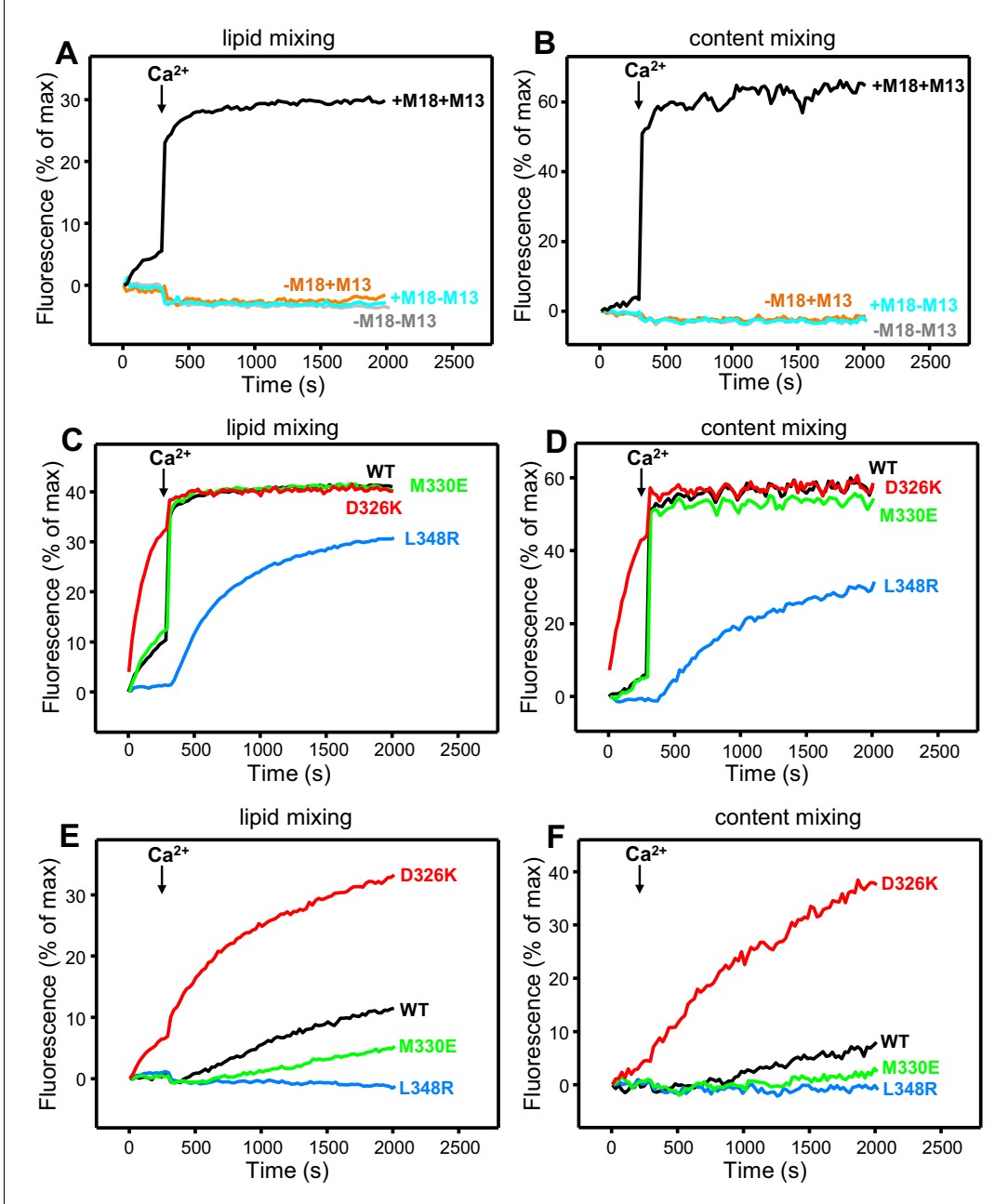

**Figure 4.** Effects of Munc18-1 mutations that alter synaptobrevin binding on membrane fusion in reconstitution assays. Lipid mixing (**A,C,E**) between V- and T-liposomes was measured from the fluorescence de-quenching of Marina Blue-labeled lipids. Content mixing (**B,D,F**) was monitored from the development of FRET between PhycoE-Biotin trapped in the T-liposomes and Cy5-Streptavidin trapped in the V-liposomes. In (**A,B**), the assays were performed in the presence of NSF-αSNAP with or without WT Munc18-1 (M18) and/or Munc13-1 $C_1C_2BMUNC_2C$ (M13) as indicated. In (**C,D**), assays were performed in the presence of NSF-αSNAP, Munc13-1 $C_1C_2BMUNC_2C$ and WT or mutant Munc18-1s as indicated. In (**E,F**), assays were performed in the presence of NSF-αSNAP, synaptotagmin-1 $C_2AB$ fragment and WT or mutant Munc18-1s as indicated. Experiments were started in the presence of 100 μM EGTA and 5 μM streptavidin, and $Ca^{2+}$ (600 μM) was added after 300 s.

The following figure supplements are available for figure 4:

**Figure supplement 1.** Effects of Munc18-1 mutations that alter synaptobrevin binding on membrane fusion in reconstitution assays.

**Figure supplement 2.** Effects of Munc18-1 mutations that alter synaptobrevin binding on membrane fusion in reconstitution assays.

from $Ca^{2+}$ binding to the $C_2B$ domain of Munc13-1 $C_1C_2BMUNC_2C$, although Synaptotagmin-1 $C_2AB$ may induce an acceleration of $Ca^{2+}$-dependent fusion that can only be measured at faster time scales (*Liu et al., 2016*). As the Munc13-1 $C_1C_2BMUNC_2C$ fragment bridges V- and T-liposomes in the absence of $Ca^{2+}$, and because this activity is key for its role in supporting membrane fusion (*Liu et al., 2016*), $Ca^{2+}$ binding to the Munc13-1 $C_2B$ domain probably helps to overcome an energy barrier to membrane fusion in these experiments. The finding that the D326K mutation designed to unfurl the Munc18-1 loop can support efficient fusion before $Ca^{2+}$ addition suggests that the furled loop conformation imposes another energy barrier in WT Munc18-1 that is tightly coupled to the barrier associated with Munc13-1 $C_1C_2BMUNC_2C$; thus, when the D326K mutation decreases the energy barrier to unfurl the Munc18-1 loop, the requirement for $Ca^{2+}$ binding to the Munc13-1 $C_2B$ domain is relaxed.

These observations led us to test the possibility that the D326K mutation may overcome, at least partially, the overall requirement for Munc13-1 $C_1C_2BMUNC_2C$ for membrane fusion. In experiments performed in the presence of NSF-αSNAP and in the absence of Munc13-1 $C_1C_2BMUNC_2C$, neither the WT nor the Munc18-1 mutants were able to support fusion between V- and T-liposomes (*Figure 4—figure supplement 2A,B*). However, when we performed analogous experiments in the presence of the Synaptotagmin-1 $C_2AB$ fragment, the D326K Munc18-1 mutant was able to stimulate fusion (*Figure 4E,F*; *Figure 4—figure supplement 2C*). This fusion was slow but was dramatically more efficient than that observed with WT Munc18-1 or the M330E and L348R mutants. These results suggest that Munc13-1 is normally critical for membrane fusion because it is needed to overcome the energy barriers, imposed by the closed conformation of syntaxin-1 and by the furled conformation of the Munc18-1 loop, to SNARE complex assembly templated by Munc18-1. Thus, the D326K mutation in Munc18-1 probably helps to overcome the need for Munc13-1 in these assays, much as a so-called LE mutation that disrupts the closed conformation of syntaxin-1 helps to overcome the requirement of Unc13 for neurotransmitter release in *C. elegans* (*Richmond et al., 2001*).

The gain-of-function associated with the D326K mutation is expected to reflect an enhanced ability to mediate SNARE complex formation. To test this notion, we used an assay that monitors SNARE complex assembly using native PAGE (*Yang et al., 2015*). Binary complexes of syntaxin-1 and WT or mutant Munc18-1s containing the M330E, D326K or L348R substitutions were formed and then incubated with synaptobrevin and SNAP-25; they were then analyzed by native PAGE. SNARE complex assembly was reflected in the appearance of a characteristic band in the middle of the gel, which was concomitant with the depletion of the band corresponding to the syntaxin-1–

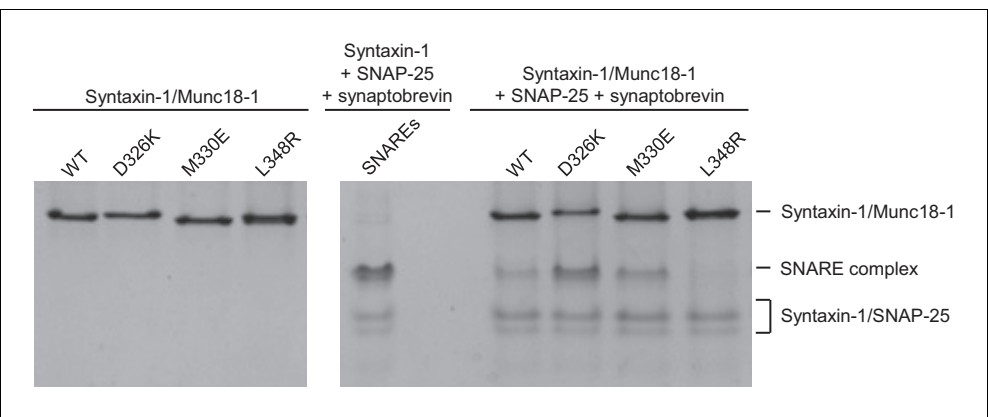

**Figure 5.** Effects of Munc18-1 mutations on SNARE complex assembly starting with the binary syntaxin-1–Munc18-1 complex. Complexes of syntaxin-1 (2–253) with WT or mutant Munc18-1 were incubated with SNAP-25 and the synaptobrevin SNARE motif, and analyzed by native PAGE followed by Coomassie blue staining (five lanes on the right). The four lanes on the left show controls with the same amounts of the syntaxin-1–Munc18-1 complexes. The middle lane shows a SNARE complex assembly reaction performed with syntaxin-1 (2–253), the synaptobrevin SNARE motif and SNAP-25 in the absence of Munc18-1. The positions of the different complexes are indicated on the right.

Munc18-1 complex at the top of the gel (*Figure 5*). Only a small amount of SNARE complex was observed when WT Munc18-1 or the M330E mutant was used, and practically no SNARE complex assembly was observed for the L348R mutant. Conversely, the D326K mutation led to a clear increase in SNARE complex formation compared to that observed with WT Munc18-1 (*Figure 5*). These results show a clear correlation between SNARE complex assembly and the activities of the mutants in the reconstitutions assays, supporting the notion that these activities are a consequence of their ability to template SNARE complex assembly.

## The D326K mutation destabilizes the structure of the Munc18-1 loop

The design of the D326K mutation was based on the furled conformation of the loop that connects two helices of domain 3a in the crystal structure of rat Munc18-1 bound to closed syntaxin-1 (*Figures 1B–D* and *6A–B*) (*Burkhardt et al., 2008*), but this loop is involved in crystal contacts that could influence the structure in this region. Similarly, the unfurled conformation of this loop in the crystal structure of Munc18-1 bound to the syntaxin-4 N-terminal peptide (*Figures 1B, D* and *6C–D*) (*Hu et al., 2011*) might have been induced by dimerization within the crystals (*Figure 6C*), and the crystal structure of squid Munc18-1 (*Bracher et al., 2000*) revealed another unfurled conformation of the loop (*Figure 6—figure supplement 1*) that could also be caused by crystal contacts. These findings show that the structure of this region of Munc18-1 is malleable and it is unclear whether the loop adopts a defined conformation in Munc18-1 alone and/or bound to syntaxin-1. Moreover, it is uncertain whether the effects of the D326K mutation on synaptobrevin binding and Munc18-1 activity arise because of structural perturbations caused by the mutation, as predicted, or from alteration of direct interactions of D326 with the SNAREs.

To shed light into these questions, we first investigated whether dimeric structures such as that observed in the Munc18-1-syntaxin-4 N-peptide crystals (*Figure 6C*) predominate in solution. Using static light scattering (SLS), we measured the following weight-average molecular mass (presented as the average of 30 determinations ± the standard deviation in kDa) at 15 μM concentration: WT Munc18-1 77.2 ± 0.1; D326K Munc18-1 78.7 ± 0.9; WT Munc18-1 bound to an unlabeled fragment spanning most of the cytoplasmic region of syntaxin-1 [syntaxin-1 (2–253)] 97.1 ± 0.6; D326K Munc18-1-syntaxin-1 (2–253) 102.1 ± 0.9. The expected molecular weights are 68.6 and 98.0 for free and syntaxin-1 (2–253)-bound Munc18-1, respectively. Hence, these data show that the free and syntaxin-1 (2–253)-bound Munc18-1 are largely monomeric under these conditions, and that the D326K mutation does not alter this behavior substantially.

We next turned to NMR spectroscopy. Clearly, it is very difficult to obtain detailed structural information on a small region of a 68-kDa protein such as Munc18-1 by any available technique in solution, but methyl TROSY-based heteronuclear multiple quantum coherence (HMQC) spectra of perdeuterated proteins that are specifically $^1$H,$^{13}$C-labeled at Ile, Leu, Met and Val methyl groups ($^2$H-ILMV-$^{13}$CH$_3$-labeling) offers the possibility of observing cross-peaks from methyl groups of very large proteins with high sensitivity (*Ruschak and Kay, 2010*), thus providing a powerful tool to investigate structural perturbations around these methyl groups. Application of this approach to Munc18-1 was hindered by its limited solubility and stability, and attempts to obtain $^2$H-ILMV-$^{13}$CH$_3$-labeled Munc18-1 yielded very low amounts of soluble protein. We circumvented this problem by preparing ILMV-$^{13}$CH$_3$-labeled Munc18-1 with only 50% perdeuteration (50%-$^2$H-ILMV-$^{13}$CH$_3$-Munc18-1), which we could obtain with reasonable yields in stable form. The quality of the $^1$H-$^{13}$C HMQC spectra obtained for 15 μM 50%-$^2$H-ILMV-$^{13}$CH$_3$-Munc18-1 in isolation or bound to syntaxin-1 (2–253) was rather limited in the Ile and Leu-Val methyl regions but, fortunately, multiple well-resolved cross-peaks could be observed with sufficient sensitivity in the Met methyl region (*Figure 6E*) and the domain 3a loop contains four methionines (M316, M324, M330 and M334; *Figure 6B*). Hence, the methyl cross-peaks from these methionines might provide information on the structural perturbations caused by the D326K mutation.

Comparison of $^1$H-$^{13}$C HMQC spectra of 50%-$^2$H-ILMV-$^{13}$CH$_3$-Munc18-1 in a free state and bound to syntaxin-1 (2–253) shows that syntaxin-1 binding induces shifts in many methionine methyl cross-peaks (*Figure 6E*), consistent with the extensive surface involved in the interaction (*Misura et al., 2000*). The $^1$H-$^{13}$C HMQC spectra acquired for D326K mutant 50%-$^2$H-ILMV-$^{13}$CH$_3$-Munc18-1 free and bound to syntaxin-1 (2–253) revealed similar changes (*Figure 6F*) to those observed for WT, as expected because this mutant retains very high affinity for syntaxin-1 (2–253) (*Table 1*) and the mutation is not in the binding interface. Attempts to assign the cross-peaks from the four methionines in

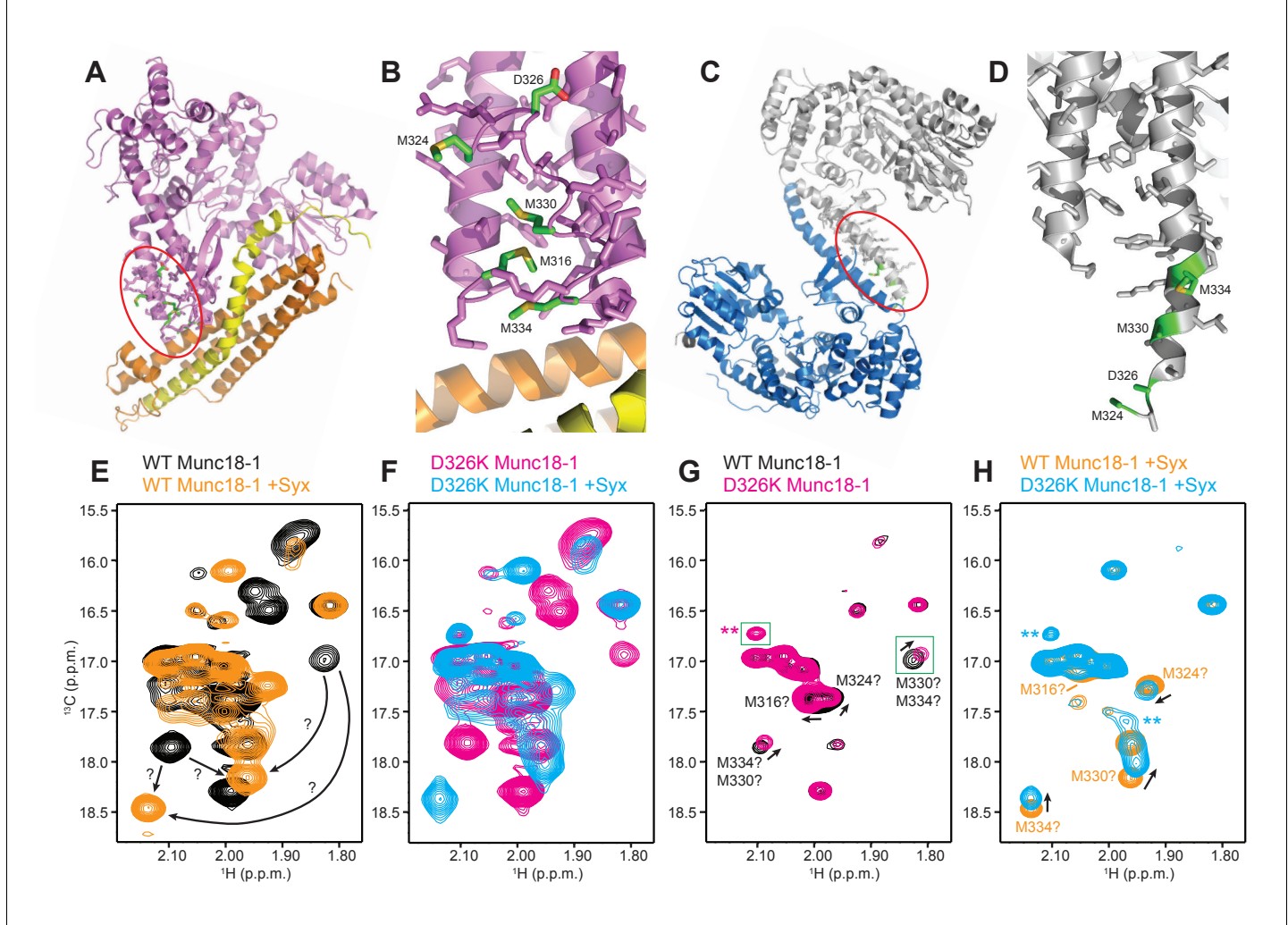

**Figure 6.** The D326K mutation destabilizes the structure of the Munc18-1 loop. (**A**) Ribbon diagram of Munc18-1 (violet) bound to syntaxin-1 (SNARE motif in yellow; N-terminal region in orange) (PDB code 3C98) (**Burkhardt et al., 2008**). The red ellipse shows the location of the loop that connects two helices of domain 3a of Munc18-1. (**B**) Close-up view of the domain 3a loop region from panel **A**. Side chains from the loop and the two helices are shown as stick models, and the atoms of the side chains from D326, M316, M324, M330 and M334 are color-coded (carbon green; oxygen red; sulfur yellow). The diagram illustrates that these side chains are well packed within the furled structure of the loop, although they have different degrees of solvent accessibility. (**C**) Ribbon diagram showing the dimeric structure observed in the crystals of Munc18-1 bound to a syntaxin-4 N-terminal peptide (not shown) (PDB code 3PUJ) (**Hu et al., 2011**). One Munc18-1 molecules is shown in gray and the other in blue. Note that dimerization involves the two domain 3a helices but the loop is at the end of the dimerization interface. The red ellipse shows the location of the loop for the gray molecule. (**D**) Close-up view of the loop in the gray Munc18-1 molecule of panel **C**. Atoms from the loop and the two helices are shown as stick models, and the atoms observed for M324, D326, M330 and M334 are color-coded (carbon green; oxygen red; sulfur yellow). Note that residues 315–323, as well as the side chains of M324, D326 and M330, were not observable, probably because they are not packed against other regions of Munc18-1 and they are thus dynamic. (**E**) Expansions from the methionine methyl region of $^1$H-$^{13}$C HMQC spectra of WT 50%-$^2$H-ILMV-$^{13}$CH$_3$-Munc18-1 free (black contours) and bound to syntaxin-1 (2–253) (orange contours). The arrows indicate potential shifts caused by syntaxin-1 (2–253) binding on the cross-peaks tentatively assigned to M330 and M334 (**Figure 6—figure supplement 2**). (**F**) Expansions from the methionine methyl region of $^1$H-$^{13}$C HMQC spectra of D326K 50%-$^2$H-ILMV-$^{13}$CH$_3$-Munc18-1 free (magenta contours) and bound to syntaxin-1 (2–253) (cyan contours). (**G,H**) Expansions of the $^1$H-$^{13}$C HMQC spectra shown in panels **E**, **F** but superimposing the spectra of WT and D326K 50%-$^2$H-ILMV-$^{13}$CH$_3$-Munc18-1 in isolation (**G**) or bound to syntaxin-1 (2–253) (**H**). The spectra were plotted at higher contour levels to emphasize the spectral changes induced by the D326K mutation. Arrows indicate cross-peak shifts. Cross-peaks are labeled with residue assignments based on spectra acquired on methionine mutants (**Figure 6—figure supplement 2**); the ? symbols after the residue numbers indicate the tentative nature of the assignments. Green boxes indicate regions of the spectra that were plotted at lower levels to show weak cross-peaks. New cross-peaks caused by the D326K mutation are indicated by **.

The following figure supplements are available for figure 6:

**Figure supplement 1.** Structure of squid Munc18-1.

*Figure 6 continued on next page*

*Figure 6 continued*

**Figure supplement 2.** Assignment of methionine cross-peaks in $^1$H-$^{13}$C HMQC spectra of 50%-$^2$H-ILMV-$^{13}$CH$_3$-Munc18-1.

**Figure supplement 3.** The D326K mutation causes selective decreases in the intensities of the $^1$H-$^{13}$C HMQC cross-peaks of methionines from the domain 3a loop of Munc18-1.

the loop by acquiring $^1$H-$^{13}$C HMQC spectra of three mutants containing single methionine substitutions (M316L, M330E and M334L) did not yield unambiguous assignments because the mutations perturbed several cross-peaks for both free and syntaxin-1 (2–253)-bound 50%-$^2$H-ILMV-$^{13}$CH$_3$-Munc18-1 (*Figure 6—figure supplement 2*). Nevertheless, the perturbations did identify four cross-peaks that are most perturbed by the mutations and hence are most likely to correspond to the M316, M324, M330 and M334 methyl groups for both the free and the syntaxin-1 (2–253)-bound 50%-$^2$H-ILMV-$^{13}$CH$_3$-Munc18-1 (*Figure 6—figure supplement 2*). Reassuringly, superposition of the $^1$H-$^{13}$C HMQC spectra obtained for WT and D326K mutant 50%-$^2$H-ILMV-$^{13}$CH$_3$-Munc18-1 revealed that the D326K mutation caused shifts in the four cross-peaks corresponding to M316, M324, M330 and M334 for both the free and the syntaxin-1 (2–253)-bound forms (*Figure 6G,H*).

The observed cross-peak perturbations caused by the D326K mutation and the three methionine mutations show that the loop region adopts well-defined structures in both isolated Munc18-1 and its complex with syntaxin-1 (2–253). The perturbations are not compatible with unfurled conformations such as that observed in the crystal structure of Munc18-1 bound to the syntaxin-4 N-terminal peptide (*Figure 6D*) because the D326, M316, M324 and M330 side chains are not well packed and the loop region is rather flexible in this dimeric structure (residues 315–323, were not observable, and the side chains of M324, D326 and M330 were also not observable). As Munc18-1 and its complex with syntaxin-1 (2–253) are largely monomeric under our conditions, additional flexibility would be expected if an unfurled loop structure were present. Conversely, the observed cross-peak perturbations are compatible with the furled loop structure observed in the crystals of Mun18-1 bound to closed syntaxin-1 (*Figure 6B*). Nevertheless, the extent to which the observed structure was influenced by crystal contacts is still unclear, and the large changes caused by syntaxin-1 (2–253) binding on the cross-peaks tentatively assigned to the M330 and M334 (*Figure 6E*) indicate that binding may substantially alter the structure of the loop. The shifts of the tentatively assigned Met cross-peaks caused by the D326K mutation (*Figure 6G,H*) show that the mutation causes some structural perturbations in the region, but these perturbations are unlikely to underlie the functional effects of the mutation because the M330E mutation induces more substantial cross-peak shifts without having functional effects.

It is noteworthy that the D326K mutation caused not only shifts but also decreased intensities in the cross-peaks from the four methionines of the Munc18-1 loop in the $^1$H-$^{13}$C HMQC spectra of the syntaxin-1 (2–253)-bound form, whereas other methionine cross-peaks generally had similar intensities in the WT and D326K spectra (*Figure 6—figure supplement 3*, right panel). Moreover, two new cross-peaks were observed in the spectrum of D326K 50%-$^2$H-ILMV-$^{13}$CH$_3$-Munc18-1 bound to syntaxin-1 (2–253) that were not present in the spectrum of the WT complex (indicated by ** in *Figure 6H*). These observations suggest that the D326K mutation destabilizes the furled loop structure, inducing the formation of an alternative (perhaps unfurled) structure in a population of the Munc18-1-syntaxin-1 (2–253) complexes, and resulting in an equilibrium between the two structures that is relatively slow in the NMR time scale. The observation of only two new cross-peaks for the alternative structure might arise because its population is low, because of overlap with other cross-peaks and/or because exchange broadening renders some new cross-peaks unobservable. Note that exchange broadening may also contribute to the decreased intensities of the cross-peaks from the four methionines in the furled loop structure. For isolated Munc18-1, the D326K mutation also caused decreases in the intensities of the four methionine cross-peaks from the furled loop, but the decreases were less marked (*Figure 6—figure supplement 3*, left panel), and only one of the new cross-peaks was observed (** in *Figure 6G*), but with lower intensity than for the syntaxin-1 (2–253)-bound form. Thus, the D326K mutation also appears to induce the formation of an alternative structure in isolated Munc18-1 but with a lower population than in the complex with syntaxin-1 (2–253). It

is tempting to speculate from these findings that formation of the alternative structure underlies the increased activity of the Munc18-1 D326K mutant in the membrane-fusion and SNARE-complex-assembly assays, as well as its increased binding to synaptobrevin, and that this increased binding is less pronounced than the increases in activity because formation of the alternative structure is less favored in isolated Munc18-1 than in the Munc18-1–syntaxin-1 complex.

## The Munc18-1 D326K mutant rescues *unc-18* nulls more efficiently than WT Munc18-1

Mammalian Munc18-1 rescues the paralysis phenotype of *unc-18* nulls in *C. elegans*, but the rescue is only partial (*Gengyo-Ando et al., 1996*) and rat Munc18-1 bearing a P335A mutation yields a much more efficient rescue than the WT protein (S. Park and S. Sugita, unpublished results). As the functional effects caused by the P335A mutation are probably related to those induced by the D326K mutation, we tested the ability of the D326K rat Munc18-1 mutant to rescue the paralysis phenotype observed in *C. elegans unc-18* (*e81*) nulls. Indeed, the D326K Munc18-1 mutant was much more efficient in rescuing paralysis than was WT Munc18-1 (*Figure 7A*). In addition, the behavioral enhancement of *unc-18* (*e81*) nulls expressing the mutant Munc18-1 was explained by increased synaptic transmission, which was indirectly measured as the sensitivity of *C. elegans* to aldicarb, an inhibitor of acetylcholine esterase (see Materials and methods). In the presence of aldicarb, *C. elegans* becomes paralyzed due to the persistent contraction of muscles arising from the accumulation of acetylcholine at the neuromuscular junction (*Miller et al., 1996*; *Nonet et al., 1998*; *Mahoney et al., 2006*). Consistent with the enhancement in motility, *unc18* (*e81*) nulls expressing

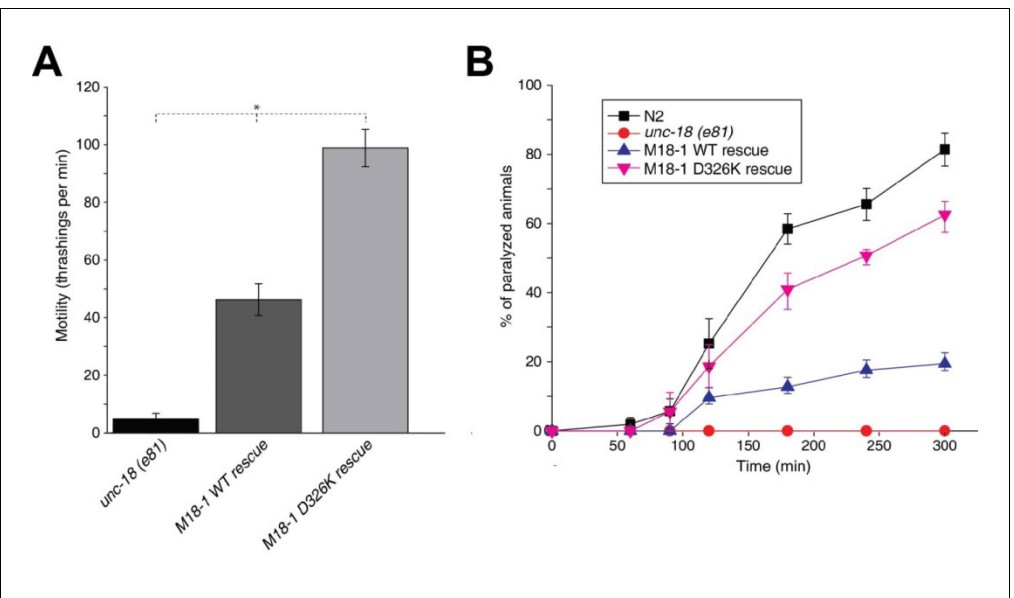

**Figure 7.** The Munc18-1 D326K mutation leads to a gain-of-function in *C. elegans*. (**A**) The motility of *C. elegans* was measured by counting the number of thrashings per min in liquid medium. *unc-18* null mutants expressing multiple copies of D326K Munc18-1 swim with higher thrashing rates than those expressing multiple copies of WT Munc18-1. Error bars indicate SEM (number n of worms per strain = 16–21). An asterisk indicates statistical significance at p<0.05. (**B**) Aldicarb assays were conducted to assess the sensitivity of rescued *unc-18* null mutants to 1 mM aldicarb. *unc-18* null mutants expressing multiple copies of D326K Munc18-1 were more sensitive to aldicarb than those expressing multiple copies of WT Munc18-1. The sensitivity of *unc-18* nulls expressing multiple copies of D326K Munc18-1 was similar to that of WT N2 worms. Error bars indicate SEM (number n of experiments = 4, where 10–20 worms per strain were analyzed in each experiment).

The following figure supplement is available for figure 7:

**Figure supplement 1.** Immunoblot analysis of *unc-18* (*e81*) mutants expressing WT and D326K mutant forms of rat Munc18-1.

the D326K mutant Munc18-1 exhibited higher sensitivity to 1 mM aldicarb than those expressing WT Munc18-1 (*Figure 7B*). Immunoblots showed comparable levels of expression for the WT and D326K mutant Munc18-1 proteins (*Figure 7—figure supplement 1*), indicating that the observed functional differences do not arise from differential expression levels. Importantly, the aldicarb sensitivity of *unc-18* (*e81*) nulls expressing the mutant Munc18-1 reached levels that were close to those of WT (N2) animals (*Figure 7B*). These experiments show that the D326K mutation in Munc18-1, which enhances synaptobrevin binding and the activity of Munc18-1 in the in vitro fusion assays, also leads to an overt gain of function in live animals.

## Discussion

Munc18-1 is crucial for neurotransmitter release much as SM proteins are crucial for traffic at most intracellular membrane compartments. The function of these proteins has proven enigmatic, in part because of the diversity of their interactions with SNAREs (*Rizo and Südhof, 2012*), but recent progress has brought key insights to help solve the mystery. Research on yeast vacuolar fusion showed that the HOPS tethering complex containing the SM protein Vps33 is required for SNARE complex assembly in the presence of Sec17 and Sec18, the yeast homologues of αSNAP and NSF (*Xu et al., 2010a*), and Munc18-1 (together with Munc13s) was later shown to orchestrate SNARE complex assembly in an NSF-αSNAP-resistant manner(*Ma et al., 2013*). The findings that a loop in domain 3a of Munc18-1 adopts distinct conformations (*Hu et al., 2011*), that Munc18-1 binds to synaptobrevin (*Xu et al., 2010b*), and that the L348R mutation that disrupts this interaction impairs Munc18-1 function, led to the notion that Munc18-1 provides a template for the formation of the SNARE complex (*Parisotto et al., 2014*). This proposal and its generality for SM protein function were emphatically supported by the crystal structures of Vps33 with Vam3 and Nyv1 (*Baker et al., 2015*). However, the dependence of Munc18-1 function on synaptobrevin binding was not clearly established, and the relation of this role of Munc18-1 to its tight functional interplay with Munc13s was unknown. Our results now reveal a clear correlation between Munc18-1 binding to synaptobrevin and its activity in stimulating membrane fusion in reconstitution assays that depend critically on both Munc18-1 and Munc13-1. Moreover, the gain-of-function observed in these assays for the D326K mutation, which was designed to unfurl the Munc18-1 loop, is mirrored in a gain-of-function in *C. elegans*. These results strongly support the template model of SM-protein function and the notion that synaptobrevin binding is autoinhibited by the furled loop conformation in domain 3a. Our data also suggest that this autoinhibition of Munc18-1 function is important to enable control of release by Munc13s, thus allowing a wide range of Munc13-dependent regulatory processes that are vital for brain function.

It may seem surprising that, although the molecular mechanism of neurotransmitter release has been studied intensely for 30 years, the importance of Munc18-1 binding to synaptobrevin has been recognized only recently. This delay arose in part because much attention was paid to the tight interaction of Munc18-1 with syntaxin-1 (*Hata et al., 1993*) and its implications in inhibiting or perhaps assisting in SNARE complex assembly (*Dulubova et al., 1999*; *Misura et al., 2000*). Munc18-1 was later found to bind to the SNARE complex and synaptobrevin binding was proposed to be important in this context (*Shen et al., 2007*; *Dulubova et al., 2007*). Binding of Munc18-1 to isolated synaptobrevin was probably missed in some early studies because it is weak and difficult to detect with the pulldown assays commonly used to identify protein–protein interactions. Recent pulldown assays did detect the interaction, but using a large synaptobrevin excess and antibody detection (*Parisotto et al., 2014*). Munc18-1–synaptobrevin binding washas been readily observed by NMR spectroscopy, which is well suited to the analysis of weak interactions (*Rizo et al., 2012*), but the interaction was dominated by the promiscuous C-terminus of the synaptobrevin SNARE motif (*Xu et al., 2010b*). Our NMR analyses now provide crucial clarification in this area.

On one hand, the findings that the C-terminus of the synaptobrevin SNARE motif also binds to the Munc13-1 MUN domain (*Figure 2—figure supplement 1*) and that its interaction with Munc18-1 is largely unaffected by the D326K or L348R mutations (*Figure 3*) strongly supports the conclusion that this interaction is not specific, although we cannot rule out its physiological relevance. On the other hand, our NMR results reveal an interaction of Munc18-1 with more central sequences of the synaptobrevin SNARE motif (*Figure 2*) that are not involved in binding to the Munc13-1 MUN domain and that, on the basis of the Vps33-Nyv1 crystal structure, are expected to be involved in

binding to Munc18-1. Moreover, Munc18-1 binding to these sequences is specifically altered by the D326K and L348R mutations (*Figure 3*). The D326K mutation designed to destabilize the furled loop conformation of domain 3a of Munc18-1 increases binding, whereas the L348R mutation impairs binding as shown earlier (*Parisotto et al., 2014*) and in agreement with the Vps33-Nyv1 binding mode (*Baker et al., 2015*). In correlation with these findings, the L348R mutation impairs Munc18-1 function in reconstitution assays and the D326K mutation stimulates Munc18-1 activity (*Figure 4*), results that also correlate with the ability of these mutants to support SNARE complex assembly starting from their binary complex with syntaxin-1 (*Figure 5*). Moreover, the D326K mutation leads to a dramatic gain-of-function in rescue experiments in *C. elegans* (*Figure 7*). Together with previous studies (*Parisotto et al., 2014*; *Baker et al., 2015*), these results strongly support the notion that Munc18-1 templates SNARE complex assembly via interactions with synaptobrevin and syntaxin-1 that resemble those of Vps33 with Nyv1 and Vam3, and that are likely to be conserved across the SM protein and SNARE families.

Establishing definitively whether and how the D326K mutation in Munc18-1 alters the structure of the domain 3a loop is challenging, but our NMR data (*Figure 6* and *Figure 6—figure supplement 2*) do show that the loop adopts defined structures that do not appear to be unfurled in both isolated and syntaxin-1 (2–253)-bound Munc18-1. The data are consistent with the furled loop structure observed in the crystal structure of the Munc18-1-syntaxin-1 complex, but indicate that the structure differs considerably in the free and syntaxin-1-bound Munc18-1. The decreases in the intensities of the methionine methyl cross-peaks from the loop region observed in the $^1$H-$^{13}$C HMQC spectra (*Figure 6—figure supplement 3*) and the appearance of new cross-peaks caused by the D326K mutation (*Figure 6G,H*) suggest that the mutation destabilizes the structure of the loop both in the isolated Munc18-1 and in the Munc18-1-syntaxin-1 (2–253) complex, but more markedly in the latter. Thus, although these results are not definitive, they support the notion that the effects of the D326K mutation on synaptobrevin binding and on the activity of Munc18-1 in the membrane fusion and SNARE complex assembly assays arise from structural destabilization, and hence that Munc18-1 is autoinhibited both in isolation and when bound to syntaxin-1.

Such autoinhibition may underlie in part the weak affinity of the Munc18-1–synaptobrevin interaction, but it appears that the interaction is intrinsically weak, as binding is not saturated at 14.5 µM protein concentration even for the D326K mutant (*Figure 2*). However, when a synaptic vesicle docks at the active zone, or when V- and T-liposomes dock in our reconstitution assays, binding should be favored by the high local concentrations of not only synaptobrevin but also Munc18-1, which initially is most likely to be bound to syntaxin-1 (*Ma et al., 2013*). Note also that weak interactions can have dramatic effects in catalyzing the assembly of a protein complex, as a ten-fold acceleration in assembly rate requires a decrease in the energy barrier of just 1.4 kcal/mol, which can be provided by a single hydrogen bond. Hence, the Munc18-1–synaptobrevin interaction might play a catalytic role in promoting SNARE complex assembly by placing the synaptobrevin and syntaxin-1 SNARE motifs near each other, in the correct register and orientation (as suggested by the Vps33 crystal structures with Nyv1 and Vam3 [*Baker et al., 2015*]). At the same time, these interactions may prevent antiparallel interactions of synaptobrevin with syntaxin-1 (see [*Weninger et al., 2003*]) that could inhibit synaptic vesicle fusion. This model does not rule out the notion that Munc18-1 may remain bound to the SNARE complex after assembly and might participate in downstream events that lead to membrane fusion. In this context, it is worth noting that the D326K mutation may increase the binding of Munc18-1 to the SNARE complex (*Table 1*), which might also underlie in part the gain-of-function caused by the mutation.

The tight $Ca^{2+}$-dependence of membrane fusion in our reconstitution assays depends on $Ca^{2+}$ binding to the Munc13-1 $C_2B$ domain by a mechanism that is not understood but that may be related to a $Ca^{2+}$-sensing role in release or promotion of an activated state of Munc13-1 during repetitive stimulation (*Liu et al., 2016*; *Shin et al., 2010*). Regardless of which of these two possibilities is correct, the facts that $Ca^{2+}$-free Munc13-1 $C_1C_2BMUNC_2C$ can dock V-liposomes to T-liposomes and that there can be efficient lipid mixing with little content mixing before $Ca^{2+}$ addition in our fusion assays indicate that Munc13-1 $C_1C_2BMUNC_2C$ helps in the fusion reaction by promoting such docking (*Liu et al., 2016*; *Xu et al., 2017*), but also imposes an energy barrier that is overcome by $Ca^{2+}$ binding to its $C_2B$ domain. In reconstitution experiments with WT Munc18-1, the additional energy barriers caused by the closed conformation of syntaxin-1 and the furled loop conformation of Munc18-1 probably hinder SNARE complex assembly to such an extent that full assembly and

membrane fusion can only occur when $Ca^{2+}$ binding to the Munc13-1 $C_2B$ domain lowers one of the existing energy barriers. Our reconstitution data show that this $Ca^{2+}$ binding event becomes less critical when Munc18-1 bears the D326K mutation (*Figure 4C,D*), probably because this mutation decreases the energy barrier caused by the furled loop conformation. In fact, some fusion can occur even in the absence of Munc13-1 $C_1C_2BMUNC_2C$ when using the Munc18-1 D326K mutant (*Figure 4E,F*), suggesting that, without a furled loop conformation in Munc18-1, SNARE complex formation can occur without an absolute requirement for the membrane bridging and syntaxin-1 opening activities of Munc13-1 $C_1C_2BMUNC_2C$.

This notion is reminiscent of the finding that syntaxin-1 bearing an LE mutation that disrupts its closed conformation (*Dulubova et al., 1999*) can partially rescue neurotransmitter release in *unc-13* nulls in *C. elegans* (*Richmond et al., 2001*), and suggests that there are at least two energy barriers within the Munc18-1–syntaxin-1 complex that hinder SNARE complex assembly and thus contribute to make Munc13-1 critical for release: one arises from autoinhibition of syntaxin-1 and the other from autoinhibition of Munc18-1. Without these autoinibitory interactions, Munc13-1 would not be essential, but then the varied modes of regulation of release during presynaptic plasticity that depend on Munc13-1 would be lost. These ideas emphasize the importance of autoinhibitory interactions in achieving the exquisite regulation of neurotransmitter release and need to be explored with further research in the future.

## Materials and

### Recombinant proteins

We previously described expression in BL21 *Escherichia coli* cells and purification of the following proteins: rat syntaxin 1A (residues 2–253); rat synaptobrevin 2 (residues 1–96 or 29–93); rat SNAP-25A full length (with the four cysteines mutated to serines); and fragments corresponding its SNARE motifs (residues 11–82 and residues 141–203); full-length rat Munc18-1; rat synaptotagmin-1 $C_2AB$ fragment (residues 131–421); full length *Cricetulus griseus* NSF V155M mutant; and full-length *Bos taurus* αSNAP (*Dulubova et al., 1999*; *2007*; *Chen et al., 2002*, *2006*; *Xu et al., 2013*; *Brewer et al., 2015*; *Ma et al., 2013*). Starting from the full-length rat Munc18-1 sequence, we used standard site-directed mutagenesis techniques and custom-designed primers to generate Munc18-1 mutants (M316L, D326K, M330E, L348R, and M334L). Mutants were expressed and purified as WT Munc18-1. Rat Munc13-1 $C_1C_2BMUNC_2C$ fragment (residues 529–1735, with the sequence from a flexible loops corresponding to residues 1408–1452 replaced by EF) was expressed and purified from Sf9 cells as described previously (*Liu et al., 2016*). To obtain uniformly $^{15}N$-labeled synaptobrevin, we used M9 minimal expression media with $^{15}NH_4Cl$ as the sole nitrogen source (1 g/L). WT and mutant 50%-$^2H$-ILMV-$^{13}CH_3$-Munc18-1 proteins were produced using M9 expression media in 50% $D_2O$ with $^2H$-glucose as the sole carbon source (3 g/L) and $^{15}NH_4Cl$ as the sole nitrogen source (1 g/L), and by adding [3,3-$^2H_2$] $^{13}C$-methyl alpha-ketobutyric acid (80 mg/L), [3-$^2H$] $^{13}C$-dimethyl alpha-ketoisovaleric acid (80 mg/L), and $^{13}C$-methyl methionine (250 mg/L) (Cambridge Isotope Laboratories, Tewksbury, Massachussetts) to the cell cultures 30 min prior to induction with 0.4 mM isopropyl $β$-D-1-thiogalactopyranoside overnight at 20°C.

### NMR spectroscopy

NMR spectra were acquired on Agilent DD2 spectrometers operating at 800 or 600 MHz. $^1H$-$^{15}N$ HSQC spectra (*Zhang et al., 1994*) were acquired at 20°C. Samples contained 14.5 μM $^{15}N$-synaptobrevin (1–96) with or without 14.5 μM Munc18-1 (WT or mutant versions) dissolved in 20 mM HEPES (pH 7.2), 120 mM KCl an 1 mM TCEP, containing 5% $D_2O$. $^1H$-$^{13}C$ HMQC spectra (*Ruschak and Kay, 2010*) were acquired at 25°C. Samples contained WT or mutant 50%-$^2H$-ILMV-$^{13}CH_3$-Munc18-1 (15–20 μM) in the absence or presence of excess syntaxin-1 (2–253) (20–22 μM), and were dissolved in 50 mM Tris pH 8.0, 150 mM KCl, 2.5 mM $CaCl_2$ and 1 mM TCEP containing 8% $D_2O$. Total acquisition times ranged from 1.5 to 6 hr for $^1H$-$^{15}N$ HSQC spectra and from 48 to 60 hr for $^1H$-$^{13}C$ HMQC spectra. All data were processed with NMRpipe (*Delaglio et al., 1995*) and analyzed with NMRView (*Johnson and Blevins, 1994*).

## Isothermal titration calorimetry

ITC experiments were performed using a VP-ITC system (MicroCal, Westborough, Massachusetts) at 20°C. Syntaxin-1 (2–253) (8–24 µM) was titrated into the cell containing Munc18-1 or mutants (0.5–2 µM) in buffer A (20 mM Hepes, pH 7.2, 120 mM KCl and 1 mM TCEP). SNARE complex (30–50 µM) was titrated into Munc18-1 or mutants (1.5–2.5 uM) in buffer B (20 mM Hepes, pH 7.4, 150 mM KCl and 1 mM TCEP). SNARE complexes were formed with SNAP-25 (11–82), SNAP-25 (141–203), syntaxin-1(2–253) and synaptobrevin (1–96). Complex assembly was accomplished by incubating a mixture of the purified fragments overnight and then removing remaining unassembled fragments by repeated rounds of size-exclusion gel filtration to efficiently eliminate unassembled syntaxin-1 (2–253), which binds very tightly to Munc18-1 and could thus perturb the measurements of SNARE complex binding. All proteins and the SNARE complex were purified by gel filtration and dialyzed in the same buffer before the experiments. The data were baseline corrected with NITPIC, fitted with a nonlinear least squares routine using a single-site binding model with ITCsy, and plotted with GUSSI (*Brautigam et al., 2016*). The 'A + B <-> AB' model was used for the fitting, and apparent concentration errors for the cell contents were compensated for by refining a single correction factor for this parameter. The 68.3% confidence intervals were obtained using the error surface projection method. Global analysis with SEDPHAT or ITCsy was not carried out in order to capture experiment-to-experiment variation in refined parameters.

## Simultaneous lipid-mixing and content-mixing assays

Assays that simultaneously measured lipid mixing from de-quenching of the fluorescence of Marina Blue-labeled lipids and content mixing from the development of FRET between PhycoE-Biotin trapped in the T-liposomes and Cy5-Streptavidin trapped in the V-liposomes were performed as described previously (*Liu et al., 2016*). V-liposomes with synaptobrevin (protein-to-lipid ratio 1:500) contained 39% POPC, 19% DOPS, 19% POPE, 20% cholesterol, 1.5% NBD PE, and 1.5% Marine Blue PE. T-liposomes with syntaxin-1–SNAP-25 (protein-to-lipid ratio 1:800) contained 38% POPC, 18% DOPS, 20% POPE, 20% cholesterol, 2% PIP2 and 2% DAG. Liposomes were in a buffer that included 25 mM HEPES, pH7.4, 150 mM KCl, 0.5 mM TCEP and 10% glycerol (v/v). V-liposomes (0.125 mM lipids) were mixed with T-liposomes (0.25 mM lipids) with various additions of the other components in a total volume of 200 µl at final a concentration of 2.5 mM $MgCl_2$, 2 mM ATP, 0.1 mM EGTA, 5 µM streptavidin, 0.8 µM NSF, 2 µM αSNAP, 1 µM Munc18-1 (wild type or mutants), and 1 µM excess SNAP-25, with or without 0.5 µM Munc13-1 $C_1C_2BMUNC_2C$, and with or without 1 µM synaptotagmin-1 $C_2AB$. T-liposomes were first incubated with NSF, $MgCl_2$, ATP, EGTA, streptavidin, NSF, αSNAP and Munc18-1 at 37°C for 25 min, and then mixed with V-liposomes and excess SNAP-25 with or without Munc13-1 $C_1C_2BMUNC_2C$ and/or synaptotagmin-1 $C_2AB$ as indicated in the figure legends. $CaCl_2$ at final concentration of 0.6 mM was added after 300 s of the start of the reaction. The fluorescence signals were monitored with a PTI spectrofluorometer (Edison, NJ). The emission fluorescence of Marina Blue-PE at 465 nm (excitation at 370 nm) was measured to monitor lipid mixing. The emission fluorescence of Cy5–streptavidin at 670 nm was measured with excitation of PhycoE-biotin at 565 nm to monitor content mixing. All experiments were performed at 30°C. At the end of each reaction, 1% w/v β-OG was added to solubilize the liposomes, and all the lipid mixing data were normalized to the maximum fluorescence signal achieved after addition of β-OG. Control experiments without streptavidin were performed to measure the maximum Cy5 fluorescence attainable upon detergent addition.

## SNARE complex assembly assays

WT or mutant Munc18-1 (6 µM) was incubated with syntaxin-1 (2–253) (5 µM) for 20 min at room temperature. Synaptobrevin (29–93) (10 µM) and SNAP-25 (10 µM) were then added and samples were incubated at room temperature for 3 hr in 25 mM Hepes, pH 7.4, 150 mM KCl, 10% glycerol (v/v). Samples were loaded onto 15% tris-glycine native gels and run at 80 V, 4°C for 6 hr. Gels were stained with Coomassie blue and imaged on a ChemiDoc MP Imaging System (Bio-Rad Laboratories, Hercules, California).

## Static light scattering

The monomeric/oligomeric state of WT and mutant Munc18-1 proteins alone or bound to syntaxin-1 (2–253) was investigated at 25°C by static light scattering using a Wyatt DynaPro NanoStar instrument (Wyatt Technology, Santa Barbara, CA). Samples contained 15 µM protein concentrations dissolved in 50 mM Tris pH 8.0, 150 mM KCl and 1 mM TCEP. The data were analyzed using the program Dynamics version 7.5.0 (Wyatt Technology, Santa Barbara, CA).

## Rescue of *unc-18* null *C. elegans* worms by extrachromosomal arrays expressing Munc18-1 variants

pMunc18-1 (a kind gift from Dr Hitoshi Kitayama, Kyoto University) described in *Gengyo-Ando et al. (1996)* was used as template to generate the rescue constructs that express WT or D326K-mutant forms of rat Munc18-1. These plasmids contained ~3 kb of *unc-18* promoter, ~1.8 kb cDNA of Munc18-1, and ~1 kb of 3' poly(A) signal of the *unc-18* gene (*Gengyo-Ando et al., 1996*). A mixture of each Munc18-1 expression plasmid and a co-injection maker (pMyo3-RFP) was prepared such that the final concentration of each DNA was ~50 ng/ul. This mixture was injected into the gonads of young adult worms of the CB81 strain (genotype: *unc-18*(*e81*)). Three to 4 days after the injection, the F1 generation, which is the progeny of the injected worms, was screened for red fluorescence, and only RFP-positive F1 worms were singled out.

## Behavioral analysis

The motility of *C. elegans* was evaluated by counting the number of thrashings per min in liquid medium. One-day-old adult worms free of OP50 bacteria were first transferred to a non-seeded plate containing 1 ml of M9 buffer. After a 2 min adjustment period in M9 buffer, the worms were video-recorded for 1 min. A single trashing was defined as a bending at the mid body, with both head and tail pointing to the opposite direction of mid body movement. For comparison of multiple groups, one-way ANOVA (analysis of variance) followed by Tukey's range test was conducted, with a significance level of 0.05. All statistical tests were performed with OriginPro 9.0.

## Aldicarb assays

The sensitivity of *C. elegans* to aldicarb was tested by examining one-day-old adult worms on non-seeded NGM plates containing 1 mM aldicarb. During the assays, animals were assessed for paralysis at multiple time points. They were considered paralyzed if their tail did not move after their head was tapped three times.

## Western blot analysis of *C. elegans* worms expressing mammalian Munc18-1 proteins

Protein extract was prepared as described in *Weimer et al. (2003)*. Briefly, we plated ~50 RFP (co-injected marker protein)-positive *unc-18* (*e81*) mutant worms expressing WT and D326K mutant rat Munc18-1 on 6 cm nematode growth medium (NGM) plates, and cultured them while manually removing RFP-negative offspring worms. As controls, we also cultured *unc-18* (*e81*) mutant worms and N2 worms. When the plates were full of worms with little OP50 left, worms were harvested and rinsed three times with a buffer containing 360 mM sucrose, 12 mM HEPES, and a protease inhibitor cocktail (1 µg/ml pepstatin A, 1 µg/ml leupeptin, 1 µg/ml aprotinin and 0.1 mM PMSF). The worms were resuspended in around five times the volume of the buffer and frozen at −80°C until use. The defrosted worms were sonicated on ice ten times with a 5 s burst. The resulting lysate was centrifuged for 15 min to pellet the cuticle, nuclei, and other debris. After centrifugation, the supernatant (final protein concentrations were ~3–5 mg/ml) was transferred to a clean microcentrifuge tube with an equal volume of sample buffer 2X. 50 µg of samples were subjected to SDS-PAGE electrophoresis followed by immunoblotting. The expression of transgenic Munc18-1 was detected by anti-Munc18-1 monoclonal antibody (1:1000) from BD Biosciences.

## Acknowledgements

We thank Victoria Esser and Yun-Zu Pan for providing purified proteins for reconstitution experiments. The Agilent DD2 consoles of the 800 MHz spectrometer and one of the 600 MHz

spectrometers used for the research presented here were purchased with shared instrumentation grants from the NIH (S10OD018027 to JR and S10RR026461 to Michael K Rosen). Bradley Quade was supported by NIH Training Grant T32 GM008297. This work was supported by grant I-1304 from the Welch Foundation (to JR), by NIH Research Project Award R35 NS097333 (to JR), which continues worked performed under NIH grants NS037200 and NS049044 (to JR), and by Canadian Institute of Health Research MOP-130573 (to SS).

## Additional information

### Funding

| Funder | Grant reference number | Author |
|---|---|---|
| National Institutes of Health | Training Grant T32 GM008297 | Bradley Quade |
| Canadian Institutes of Health Research | MOP-130573 | Shuzo Sugita |
| National Institutes of Health | R35 NS097333 | Josep Rizo |
| Welch Foundation | I-1304 | Josep Rizo |
| National Institutes of Health | S10OD018027 | Josep Rizo |
| National Institutes of Health | NS037200 | Josep Rizo |
| National Institutes of Health | NS049044 | Josep Rizo |

The funders had no role in study design, data collection and interpretation, or the decision to submit the work for publication.

### Author contributions

ES, Conceptualization, Formal analysis, Funding acquisition, Validation, Investigation, Methodology, Writing—review and editing; JX, SP, BQ, CAB, Conceptualization, Formal analysis, Validation, Investigation, Methodology, Writing—review and editing; XL, Formal analysis, Validation, Investigation, Methodology; KSt, Conceptualization, Formal analysis, Investigation, Methodology, Writing—review and editing; KSu, Formal analysis, Investigation, Methodology; SS, Conceptualization, Formal analysis, Funding acquisition, Validation, Investigation, Methodology, Writing—original draft, Project administration, Writing—review and editing; JR, Conceptualization, Formal analysis, Supervision, Funding acquisition, Validation, Investigation, Methodology, Writing—original draft, Project administration, Writing—review and editing

### Author ORCIDs

Shuzo Sugita, http://orcid.org/0000-0002-9182-873X
Josep Rizo, http://orcid.org/0000-0003-1773-8311

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
