## [Decision Letter]

Thank you for submitting your article "Autoinhibition of Munc18 Modulates Synaptobrevin Binding and Enables Munc13-dependent Regulation of Synaptic Exocytosis" for consideration by *eLife*. Your article has been reviewed by three peer reviewers, and the evaluation has been overseen by a Reviewing Editor and Randy Schekman as the Senior Editor. The reviewers have opted to remain anonymous.

The reviewers have discussed the reviews with one another and the Reviewing Editor has drafted this decision to help you prepare a revised submission.

Summary:

A recent paper (Baker 2015) proposed, based on X-ray crystallographic studies of vacuolar proteins, that SM proteins function as templates that orient and align R- and Qa-SNAREs for assembly. This model was not, however, compatible with a previous study from the author's lab (Xu 2010) of the C terminus of the synaptobrevin SNARE motif binding to Munc18.

Here, the discrepancy is satisfyingly resolved, with new NMR data that strongly suggest that vacuolar and synaptic SM proteins bind R-SNAREs in similar ways after all. A new Munc18 mutant, D326K, is designed with the hope of destabilizing the 'closed' (previously 'furled') conformation of domain 3a relative to the 'open' conformation. [Other studies have used P335A to stabilize the open conformation relative to the closed.] This mutant binds a bit more tightly to synaptobrevin, and perhaps also to the SNARE complex, and displays strong gain-of-function features in vitro and in vivo.

The studies are impressive in both technical range and quality, and establish the importance of SM association with the SNARE domains of R-SNAREs, establishing this interaction's generality.

While this manuscript was favorably reviewed, the accompanying manuscript by Park et al. was rejected. The authors may optionally consider to incorporate some of the results from Park et al. that are most relevant for the paper by Sitarska.

Essential revisions:

1) Like previous studies (and also the accompanying manuscript), this study lacks direct evidence for the impact of the mutations (either D326K or P335A) on the conformation and/or dynamics of domain 3a. The story would be stronger if the authors could provide such evidence.

2) Do the ITC data convincingly rule out a reduction in Munc18-syntaxin affinity caused by the D326K mutation?

3) It is not convincing that the domain 3a closed conformation by itself represents a major barrier to SNARE complex assembly – the available SM structures would appear to suggest that domain 3a is flexible, although generally 'open' or 'closed' conformations are observed frequently (the latter often stabilized by crystal contacts). Might it instead be that the importance of the mutation only manifests itself within complexes that have not heretofore been defined in detail?

4) The data in *C. elegans* compare worms expressing a variable (and presumably unknown) copy number of Munc18-1 WT and D326K mutant. It would be necessary to compare the expression levels directly using Western blot or similar. If the expression levels are not similar, the data are hard to interpret. Why are none of the worms as sensitive as the N2 (wildtype) worms? Is this because the Munc18-1 expression levels are lower than in the WT case?

---

## [Author Response]

*Essential revisions:*

*1) Like previous studies (and also the accompanying manuscript), this study lacks direct evidence for the impact of the mutations (either D326K or P335A) on the conformation and/or dynamics of domain 3a. The story would be stronger if the authors could provide such evidence.*

We fully agree with this concern. Unfortunately, it is difficult to address it in a definitive manner because determining the structure in solution of a 70 kDa protein with a high tendency to aggregate is very challenging if not impossible, and the loop region is involved in crystal contacts in all the crystal structures of Munc18-1 described so far, leaving an uncertainty as to what is the structure of the loop even for the WT protein. Nevertheless, this is an important point and we have devoted a strong effort to perform NMR studies that could shed some light into this issue, which has caused the delay in submitting the revised manuscript. We have obtained NMR data that are presented in a new section (subsection “The D326K mutation destabilizes the structure of the Munc18-1 loop”) as well as in Figure 6 and Figure 6—figure supplement 2 and Figure 6—figure supplement 3. The data are not definitive because of the intrinsic difficulty of the problem and hence we have been cautious when drawing conclusions. However, the results do show that the loop is structured in both isolated Munc18-1 and Munc18-1 bound to syntaxin-1, and suggest that the D326K mutation destabilizes the loop structure in both cases, particularly in the Munc18-1-syntaxin-1 complex. Hence, the new data provide an explanation for the finding that the D326K mutation increases binding to synaptobrevin, albeit moderately, while it has strong effects on the activity of Munc18-1 in the membrane fusion and SNARE complex assembly assays, which rely on the formation of the Munc18-1-syntaxin-1 complex.

*2) Do the ITC data convincingly rule out a reduction in Munc18-syntaxin affinity caused by the D326K mutation?*

The ITC data does not rule out this possibility. In the revised manuscript, we have changed the corresponding sentence, which now reads: ‘Hence, these data show that the three Munc18-1 mutants retain very high affinity for syntaxin-1, although we cannot rule out that the mutations alter the affinity to some extent that is not measurable in these experiments.’ Nevertheless, given the consistency of all the data presented in the paper and the very high affinity that the D326K mutant has for syntaxin-1, it seems rather unlikely that the effects of the D326K mutation arise because from a reduction in the affinity for syntaxin-1.

*3) It is not convincing that the domain 3a closed conformation by itself represents a major barrier to SNARE complex assembly – the available SM structures would appear to suggest that domain 3a is flexible, although generally 'open' or 'closed' conformations are observed frequently (the latter often stabilized by crystal contacts). Might it instead be that the importance of the mutation only manifests itself within complexes that have not heretofore been defined in detail?*

This is again a very good point. However, as we discuss now in the section presenting the new NMR data, the unfurled conformations of the loop observed in two crystal structures of Munc18-1 may have been induced by crystal contacts. Moreover, the effects of methionine mutations that we made in the loop to assign 1 H- 13C HMQC cross-peaks, as well as the effects of the D326K mutation itself (Figure 6 and Figure 6—figure supplement 2), are incompatible with the unfurled structures observed in those crystal structures. Overall, the NMR data indicate that there is a defined, well-packed loop structure (hence not unfurled) in Munc18-1 that changes but remains furled upon syntaxin-1 binding.

*4) The data in C. elegans compare worms expressing a variable (and presumably unknown) copy number of Munc18-1 WT and D326K mutant. It would be necessary to compare the expression levels directly using Western blot or similar. If the expression levels are not similar, the data are hard to interpret. Why are none of the worms as sensitive as the N2 (wildtype) worms? Is this because the Munc18-1 expression levels are lower than in the WT case?*

In Figure 7—figure supplement 1, we now present a Western blot showing similar expression of WT and D326K mutant Munc18-1. The observation that mammalian Munc18-1 can partially, but not completely, rescue phenotypes of invertebrate Unc18 nulls in *C. elegans* was reported in 1996 (Gengyo-Ando et al. 1996). It is not surprising that the rescue is not complete given the evolutionary distance between worms and mammals. Note that we cannot directly compare the expression of mammalian Munc18-1 with that of Unc18 in N2 animals because the antibody used to recognize the mammalian protein does not recognize *C. elegans* Unc18 (Figure 7—figure supplement 1).